# Stroma-derived Dickkopf-1 contributes to the suppression of NK cell cytotoxicity in breast cancer

Seunghyun Lee [1], Biancamaria Ricci[1], Jennifer Tran [2], Emily Eul[1], Jiayu Ye[3], Qihao Ren [3], David Clever[1], Julia Wang [2,4], Pamela Wong [2], Michael S. Haas[5], Sheila A. Stewart [2,3,6], Cynthia X. Ma [2,6], Todd A. Fehniger [2,6] & Roberta Faccio [1,6,7] ✉

Mechanisms related to tumor evasion from NK cell-mediated immune surveillance remain enigmatic. Dickkopf-1 (DKK1) is a Wnt/β-catenin inhibitor, whose levels correlate with breast cancer progression. We find DKK1 to be expressed by tumor cells and cancer-associated fibroblasts (CAFs) in patient samples and orthotopic breast tumors, and in bone. By using genetic approaches, we find that bone-derived DKK1 contributes to the systemic DKK1 elevation in tumor-bearing female mice, while CAFs contribute to DKK1 at primary tumor site. Systemic and bone-specific DKK1 targeting reduce tumor growth. Intriguingly, deletion of CAF-derived DKK1 also limits breast cancer progression, without affecting its levels in circulation, and regardless of DKK1 expression in the tumor cells. While not directly supporting tumor proliferation, stromal-DKK1 suppresses NK cell activation and cytotoxicity by downregulating AKT/ERK/S6 phosphorylation. Importantly, increased DKK1 levels and reduced cytotoxic NK cells are detected in women with progressive breast cancer. Our findings indicate that DKK1 represents a barrier to anti-tumor immunity through suppression of NK cells.

Breast cancer is one of the most frequently diagnosed malignancies in women[1]. Despite early diagnosis, approximately 20-30% of patients experience metastatic relapse within 5-10 years of curative-intent therapy[2]. Hence, the need to find long-term and highly effective therapeutic approaches to prevent recurrence.

Immune checkpoint blockade (ICB) has been FDA-approved to treat a variety of cancers previously considered incurable. Pembrolizumab (anti-PD-1) in combination with chemotherapy is now a standard-of-care for early and advanced triple-negative breast cancer (TNBC)[3,4]. However, over 30% of TNBC patients do not benefit from ICB, and ICB is also not effective for ER+ breast cancer[5]. Key factors responsible for the poor therapeutic responses to ICB in breast cancer

include low tumor mutational burden[6], recruitment of suppressive immune cells, and exclusion of lymphocytes at the tumor sites[7]. A recent analysis of circulating immune cell populations in breast cancer patients refractory to neoadjuvant chemotherapy revealed an apparent increase in the proportion of dysfunctional CD56$^{dim}$/CD16$^-$ NK cells compared to patients achieving a pathological Complete Response (pCR)[8]. NK cells are innate immune populations offering a first line of defense against incipient tumors[9]. NK cell intratumoral and peritumoral abundance correlate with an elevated pCR rate in breast cancer patients undergoing neoadjuvant chemotherapy[10]. Furthermore, preclinical evidence suggests that activated NK cells could potentiate the response to ICB in MHC class I low tumors[11]. Thus, understanding the

[1]Department of Orthopaedic Surgery, Washington University School of Medicine, St. Louis, MO, USA. [2]Department of Medicine, Washington University School of Medicine, St. Louis, MO, USA. [3]Department of Cell Biology and Physiology, Washington University School of Medicine, St. Louis, MO, USA. [4]McDonnell Genome Institute, Washington University in St. Louis, St. Louis, MO, USA. [5]Leap Therapeutics, Cambridge, MA, USA. [6]Siteman Cancer Center, Washington University School of Medicine in St. Louis, St. Louis, MO, USA. [7]Shriners Hospitals for Children St Louis, St Louis, MO, USA. ✉e-mail: faccior@wustl.edu

mechanisms that drive systemic immunosuppression, including NK cell inactivation, could improve response to ICB in breast cancer patients.

Dickkopf-1 (DKK1) is a soluble inhibitor of the Wnt/β-catenin pathway, primarily recognized for its role in bone homeostasis and cancer-induced osteolytic bone disease[12,13]. Elevated levels of DKK1 in circulation and/or in tumor tissues correlate with poor prognosis in numerous cancer types[14–19]. Direct effects of DKK1 on tumor cells have been described in head and neck cancer, pancreatic ductal adenocarcinoma, and esophageal squamous cell carcinoma[20–22]. Our lab and others have also demonstrated that DKK1 drives the accumulation of suppressive myeloid populations in melanoma, lung carcinoma, prostate, and gastric cancer[18,23–27], ultimately reducing T cell and NK cell responses. Thus, DKK1 targeting has been successfully used in combination with ICB in mouse models of melanoma and gastric cancer[24,26,28]. In breast cancer, high levels of DKK1 have been associated with poor prognosis and dissemination to bone[15]. The pro-tumorigenic effects of DKK1 in breast cancer have been mainly attributed to its ability to increase bone resorption, thus creating a favorable environment for tumor dissemination to bone[15,29]. However, whether DKK1 exerts systemic immune suppressive effects and/or generates a local immune suppressive environment at the primary tumor site has never been reported.

In this work, we demonstrate the direct inhibitory effects of DKK1 on NK cell activation and cytotoxicity during breast cancer progression. We show that cancer-associated fibroblasts (CAFs) contribute to the production of DKK1 in the tumor microenvironment (TME), and bone cells contribute to systemic DKK1 elevation. Both bone- and CAF-derived DKK1 suppress NK cell cytotoxicity, and their targeting reduces tumor progression. Finally, increased DKK1 levels and reduced cytotoxic NK cells are also detected in breast cancer patients with progressive bone disease. Our work positions DKK1 as a negative modulator of anti-tumor immunity via suppression of NK cell cytotoxicity and raises the importance of DKK1 targeting to improve NK cell-directed therapies.

## Results

### DKK1 augments breast cancer progression

To determine the role of DKK1 in breast cancer progression, we first measured DKK1 serum levels in C57BL/6 mice orthotopically injected with a luminal B, ER⁺, hormone-resistant PyMT-BO1 cell line into the mammary fat pad (MFP). DKK1 levels were significantly increased in tumor-bearing mice (Fig. 1A), recapitulating findings in breast cancer patients[17]. Next, we administered the DKK1-neutralizing monoclonal antibody mDKN01, previously validated for its anti-tumor effects in melanoma, gastric, and gynecologic mouse tumor models[24,26,30–32] and currently under investigation in clinical trials for gastric and endometrial cancers (NCT04363801, NCT05761951, NCT04681248). Administration of mDKN01 (10 mg/kg) every other day following tumor inoculation, led to a significant reduction in primary tumor growth compared to IgG (Fig. 1B). To further investigate the role of DKK1 during tumor dissemination, we injected the firefly luciferase-conjugated PyMT-BO1 cell line into albino C57BL/6 mice either intracardiacally (i.c.) to study dissemination to various organs including the bone, or directly into the tibias (i.t.) to study tumor growth in bone, followed by treatment with mDKN01. Bioluminescence imaging (BLI) showed significantly reduced tumor growth at all sites (Fig. 1C–F). To assess the therapeutic potential of mDKN01 on established tumors, we initiated treatment 7 days post-tumor inoculation (Fig. 1G) and observed a significant reduction in tumor burden compared to IgG controls (Fig. 1H). Similarly, DKK1 neutralization significantly reduced the growth of the luminal B, ER⁺/PR⁺, hormone-sensitive EO771 breast cancer cell line (Supplementary Fig. 1A).

To assess whether DKK1 levels were also increased in TNBC, we used BALB/c mice orthotopically injected with the 4T1 tumor line and found increased DKK1 in circulation (Fig. 1I). Administration of

mDKN01 (10 mg/kg) led to a significant reduction in 4T1 tumor growth at the primary site and in the bone compared to IgG control (Fig. 1J–L). These results demonstrate the involvement of DKK1 in supporting tumor progression and the therapeutic benefit of DKK1 targeting in various breast cancer subtypes.

### DKK1 is expressed in the tumor microenvironment of breast cancer patient tissues

To assess DKK1 expression at the tumor site, we analyzed a microarray dataset (GSE3744[33]) composed of healthy and malignant breast tissues from ER⁺, HER2⁺, and TNBC patients. DKK1 expression was increased in triple-negative (TNBC, Fig. 2A) and HER2⁺ breast cancer (Fig. 2C) compared to healthy tissues, whereas no significant differences were observed in ER⁺ breast cancer (Fig. 2E). To further identify which cell populations express DKK1, we analyzed a single-cell RNAseq dataset from 26 primary human breast tumors, including 11 ER⁺, 5 HER2⁺, and 10 TNBC (GSE176078[34]). The cells were annotated and clustered by using canonical and signature-based markers[35] (Supplementary Fig. 2A, B). Confirming the microarray analysis, DKK1 was expressed in the cancer epithelial cells in HER2⁺ and TNBC, while it was detected at much lower levels in ER⁺ and in the normal epithelial cells (Supplementary Fig. 2C). No DKK1 expression was detected in the immune cell cluster.

Since DKK1 levels are increased in circulation in ER⁺ patients[36], we sought to examine whether DKK1 was expressed by additional cell types in the tumor microenvironment (TME). Interestingly, we detected DKK1 expression, albeit limited, in the stromal compartment within CAF clusters (Supplementary Fig. 2D). Further subtype analysis based on established breast cancer CAF classification[37–39], showed DKK1 expression in ACTA2⁺COL1a1^high PDGFRα⁺ myofibroblasts (myCAF, Supplementary Fig. 2E–G).

To confirm these findings, we performed automated multiplex immunohistochemistry (IHC) using TNBC, HER2⁺, and ER⁺ human breast cancer tissue microarrays (TMAs). For each antibody used, images were given an arbitrary color with the deconvolution algorithm provided by HALO software under the supervision of a trained pathologist, which allowed us to determine the populations expressing DKK1. The specificity and titration of the anti-DKK1 antibody were determined using the placenta as a positive control and benign breast epithelium as a negative control (Supplementary Fig. 2H). Placenta and normal breast tissues were included in the TMA to set up the threshold signal in HALO and eliminate any background noise (Supplementary Fig. 2I, J). As expected, DKK1 staining was observed in TNBC and HER2⁺ cancer epithelial cells (Fig. 2B, D), identified by the expression of the pan-cytokeratin (PanCK) marker, but very limited expression was found in ER⁺ breast cancer (Fig. 2F). Interestingly, the epithelial cancer cells displayed a strong DKK1 nuclear localization, as recently reported[40], rather than a cytoplasmic staining where secreted proteins typically reside. Although DKK1 is expected to be secreted, we were unable to detect a robust extracellular signal, possibly due to the technical limitations of using multiplex IHC. Diffused intracellular DKK1 staining was detected in subsets of PDGFRα⁺ and αSMA⁺ cells in TNBC and HER2⁺ subtypes (Fig. 2B, D), but it was barely measurable in the stroma of ER⁺ tumors (Fig. 2F), in agreement with a previous report[41]. We also observed that DKK1 was increased in a microarray dataset from invasive ductal carcinoma (IDC) compared to normal breast tissue (Fig. 2G; GSE8977[42]). Next, we performed multiplex IHC on breast biopsies from 13 patients diagnosed with ductal carcinoma in situ (DCIS, stage 0 breast cancer) with clinical annotations, of which 3 patients had no recurrence and 10 patients developed ipsilateral breast cancer after partial mastectomy. Interestingly, we observed DKK1 staining only in cells expressing the fibroblast markers PDGFRα and COL14a1, but not in the epithelial cells, in 9 out of 10 in DCIS tissues from patients who recurred (Fig. 2H Recurrence). DKK1 protein expression was not detected in any of the DCIS patients who did not recur (Fig. 2H No Recurrence).

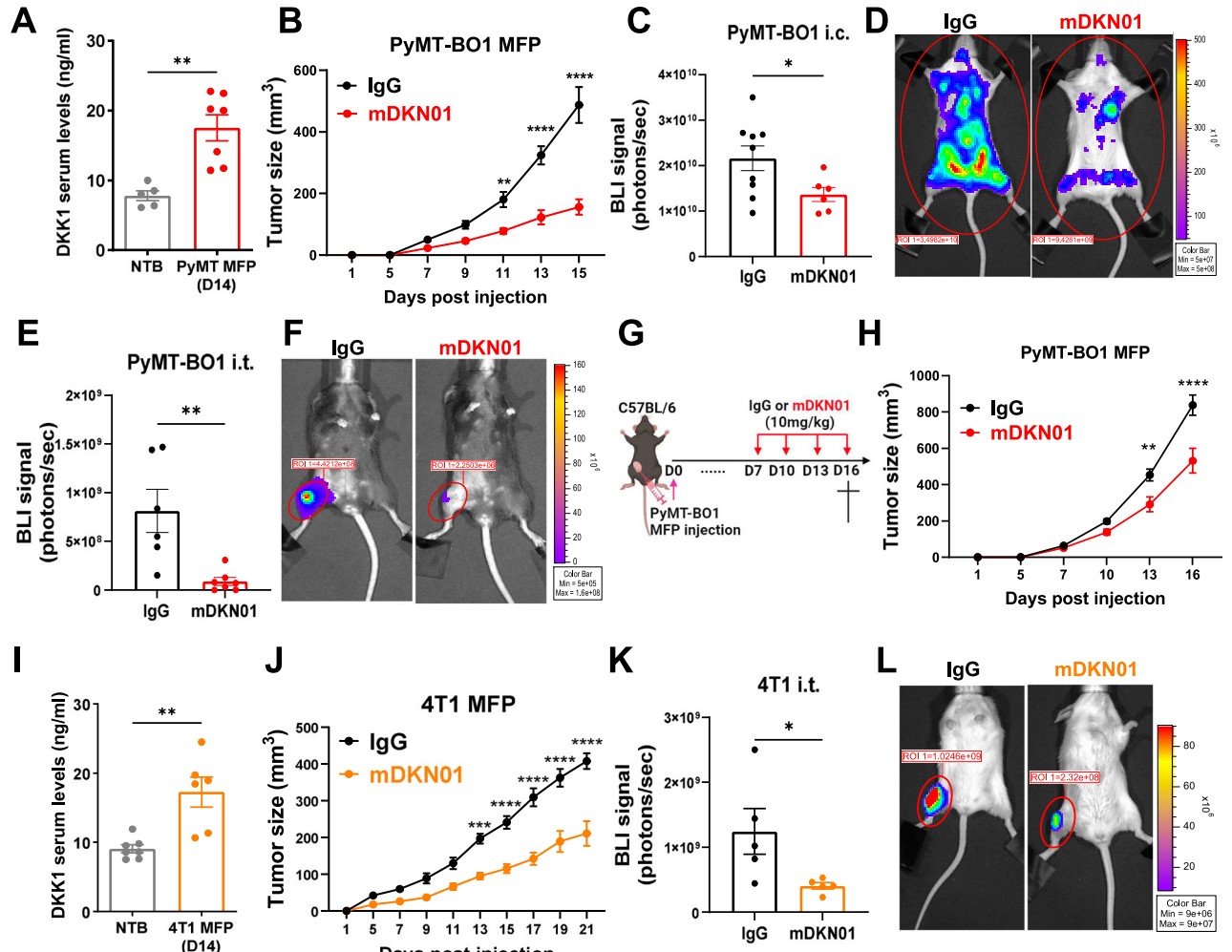

**Fig. 1 | DKK1 augments breast cancer progression. A** DKK1 serum levels were measured by ELISA in no tumor-bearing (NTB) 6–8 weeks old C57BL/6 WT female mice (n = 5) or 2 weeks after the inoculation of PyMT-BO1 breast cancer cells into the mammary fat pad (MFP; 10⁵ cells, n = 7). **B** Tumor growth in the MFP was determined by caliper measurements in WT mice inoculated with PyMT-BO1 (n = 5 mice/group) receiving mDKN01 (10 mg/kg) or control IgG antibody i.p. every other day. **C–F** Tumor progression was determined by BLI in mice inoculated intra-cardiacally with PyMT-BO1 cells (i.c.; 10⁴ cells, albino C57BL/6, n = 9 for IgG, n = 6 for mDKN01) (**C, D**) or intratibially (i.t.; 10⁴ cells, C57BL/6, n = 6 for IgG, n = 7 for mDKN01) (**E, F**) followed by administration of mDKN01 or control IgG antibody every other day. **G** Schematic representation of the therapeutic administration of IgG and mDKN01. **H** Tumor growth in the MFP was determined by caliper measurements in WT mice inoculated with PyMT-BO1 (n = 5 mice/group) receiving

mDKN01 (10 mg/kg) or control IgG antibody i.p. every three days starting 7 days post-tumor inoculation. **I** DKK1 serum levels were measured by ELISA in 6–8 weeks old female BALB/c WT mice with no tumors (NTB n = 7) or 2 weeks after the inoculation of 4T1 breast cancer cells into the MFP (n = 6). **J** Primary tumor growth was evaluated by caliper measurements in WT mice inoculated with 4T1 cells (n = 4 mice/group) into the MFP receiving mDKN01 (10 mg/kg) or control IgG antibody i.p. every other day. **K, L** Tumor progression was determined by BLI in mice inoculated intratibially with 4T1 cells (i.t.; 10⁴ cells, BALB/c, n = 5/group) followed by administration of mDKN01 or control IgG antibody every other day. Results are shown as mean ± SEM. An unpaired t-test with a two-tailed P-value (**A, C, E, I, K**), and two-way ANOVA followed by Bonferroni multiple-comparison test (**B, H, J**) were used to determine significance. *P < 0.05, **P < 0.01, ***P < 0.001, ****P < 0.0001. Source data are provided as a Source Data file.

## DKK1 is expressed in cancer-associated fibroblasts and bone in murine breast cancer models

Next, we investigated *Dkk1* expression in the PyMT, 4T1, and E0771 murine tumors. While *Dkk1* transcripts were only detected in the 4T1 tumor cell line in vitro (Supplementary Fig. 1B), *Dkk1* was found in the tumor mass of PyMT and E0771 models (Supplementary Fig. 1C, D). IHC showed DKK1 staining in stromal cells with an elongated, fibroblast-like morphology in the PyMT (Fig. 2I), spontaneous MMTV-PyMT (Fig. 2J), and 4T1 orthotopic tumors (Fig. 2K). Further, co-staining with CAF and epithelial markers confirmed DKK1 in elongated αSMA⁺ cells and partial colocalization with COL1a1 (Fig. 2L, M, Supplementary Fig. 2K). As expected, DKK1 was also detected in the 4T1 tumor cells, but not in the PyMT cancer cells.

Because DKK1 levels in circulation are increased in ER⁺ patients with bone metastases[36] and *Dkk1* is highly expressed in bone in

homeostatic conditions[43], we evaluated the expression of bone-derived *Dkk1* in mice bearing orthotopic breast tumors and found increased expression compared to no tumor controls (Supplementary Fig. 1E, F). These results indicate local production of *Dkk1* at the tumor site by either tumor cells and/or CAFs, and distal production of *Dkk1* by bone cells.

## Bone and CAF-derived DKK1 contribute to systemic and local increases in DKK1 levels during tumor progression

To determine the role of bone versus CAF-derived DKK1 during tumor progression, we generated mouse models with targeted deletion of *Dkk1* in osteoblasts and fibroblasts. To specifically delete *Dkk1* from osteoblasts, we crossed *Dkk1*^fl/fl^ mice with the doxycycline-repressible Sp7 Cre line (herein referred to as Sp7-*Dkk1*cKO)[44]. Because *Dkk1* deletion leads to embryonic lethality, moms and pups were fed a

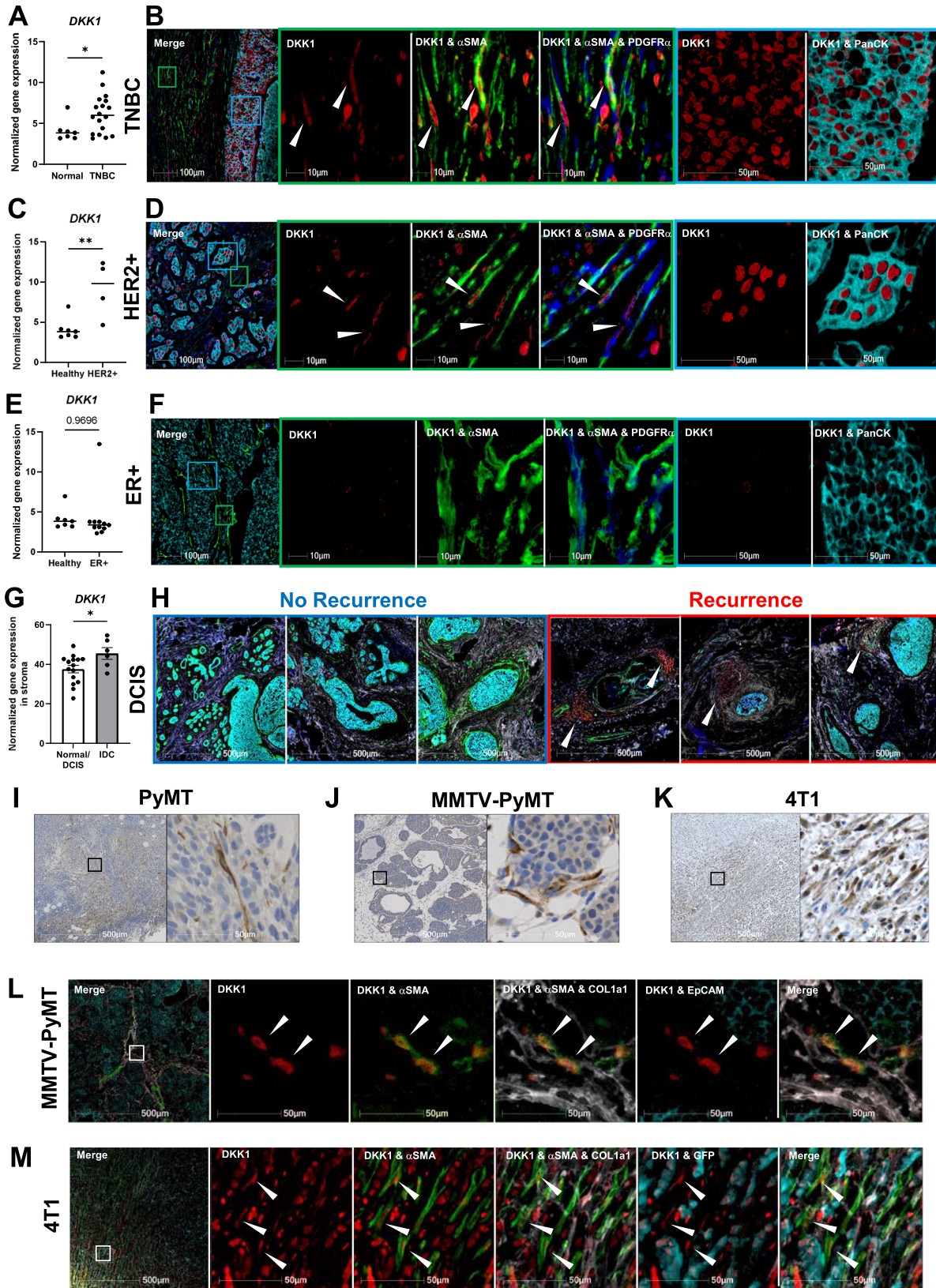

doxycycline-containing diet until weaning to suppress the transgene activation, and mice were orthotopically injected with the PyMT tumor cells when they reached 6–8 weeks of age. Strikingly, Sp7-*Dkk1*cKO mice showed a significant reduction in primary tumor growth (Fig. 3A). While *Dkk1* expression at the tumor site was not reduced compared to littermate controls, DKK1 levels in circulation were drastically reduced,

indicating that bone-derived DKK1 contributes to systemic elevation of DKK1 during tumor progression (Fig. 3B-D).

Based on the expression of DKK1 in αSMA[+] cells in the TME, next, we crossed *Dkk1*[fl/fl] mice with the inducible αSMACreER[T2] line (referred to as αSMA-*Dkk1*cKO). Cre activation was induced by 5 consecutive intraperitoneal (i.p.) injections of tamoxifen (100 mg/kg per dose) to

**Fig. 2 | DKK1 is expressed in the tumor microenvironment. A, C, E** Normalized gene expression of DKK1 in healthy human breast tissues (*n* = 7) and triple-negative breast cancer (TNBC) (**A**, *n* = 18), HER2⁺ breast cancer (**C**, *n* = 4), ER⁺ breast cancer (**E**, *n* = 11) (GSE3744). **B, D, F** Representative images of multiplex immunohistochemistry (IHC) of TNBC (**B**, *n* = 20), HER2⁺ (**D**, *n* = 12), and ER⁺ (**F**, *n* = 8) human breast cancer subtypes stained for DKK1 (red), αSMA (green), PDGFRα (blue), and panCK (cyan). The green inset highlights the stromal area and the cyan inset highlights the tumor area. **G** Normalized gene expression of DKK1 in the stroma derived from normal breast tissue or invasive ductal carcinoma (IDC) (GSE8977). Results are shown as mean ± SEM. **H** Representative images of multiplex IHC of human ductal carcinoma in situ (DCIS) samples (*n* = 13) stained for DKK1 (red),

αSMA (green), PDGFRα (blue), COL14a1 (white), and panCK (cyan) from patients who did not develop ipsilateral breast cancer (blue box) versus patients who developed ipsilateral breast cancer (red box). **I–K** Representative images of DKK1 staining by IHC (brown) in orthotopic PyMT tumors (**I**, *n* = 5), spontaneous MMTV-PyMT breast tumors (**J**, *n* = 5), and orthotopic 4T1 tumors (**K**, *n* = 4). **L, M** Representative images of multiplex IHC of spontaneous MMTV-PyMT breast tumors (**L**, *n* = 5), and orthotopic 4T1 tumors (**M**, *n* = 4) stained for DKK1 (red), αSMA (green), COL1a1 (white), and EpCAM (MMTV-PyMT tumor cells, cyan) or GFP (4T1 tumor cells, cyan). An unpaired *t*-test with a two-tailed *P*-value (**A, C, E, G**) was used to determine significance. Source data are provided as a Source Data file.

10–12 weeks old αSMA-*Dkk1*cKO and αSMA-*Dkk1*WT mice. As a control, we crossed the αSMACreER^T2 mice with the Rosa26-LSL-tdTomato line (referred to as αSMA-tdT mice) and injected PyMT cells into their MFP one day after the first tamoxifen injection to confirm the presence of αSMA-tdT⁺ cells exclusively at the tumor site, but not in the bone (Supplementary Fig. 3A). Intriguingly, αSMA-*Dkk1*cKO mice also showed a significant reduction in primary tumor growth compared to their littermate controls (Fig. 3E), despite displaying no changes in DKK1 levels in the bone and circulation but achieving efficient deletion in the tumor mass (Fig. 3F–H). Multiplex IHC further confirmed expression of DKK1 or lack thereof in αSMA⁺ cells (Fig. 3I). Confirming the estrogen insensitivity of the PyMT line, tamoxifen did not directly affect tumor cell proliferation in vitro (Supplementary Fig. 3B) and PyMT subcutaneous (SQ) tumor growth was reduced in tamoxifen-treated αSMA-*Dkk1*cKO male mice compared to controls (Supplementary Fig. 3C).

Because αSMA is also expressed in myoepithelial and endothelial cells, to further determine the relevance of CAF-derived DKK1 during tumor progression, as a complementary approach we crossed *Dkk1*^fl/fl mice with the FSP1 Cre line (referred to as FSP1-*Dkk1*cKO) to allow deletion of *Dkk1* in fibroblasts. Similarly to the αSMA-*Dkk1*cKO mice, we observed a significant reduction in primary tumor growth in this model (Supplementary Fig. 3D). Efficient deletion of *Dkk1* in the TME was confirmed via qRT-PCR (Supplementary Fig. 3E).

Next, to further understand the local effects of DKK1 in the TME, we isolated tdT⁺ CAFs from primary tumors in αSMA-*Dkk1*cKO-tdT and αSMA-*Dkk1*WT-tdT mice and co-injected them with PyMT tumor cells (1:1 ratio) into the MFP of naïve WT recipient mice. Expression or deletion of *Dkk1* in the sorted tdT⁺ CAFs was confirmed by qRT-PCR (Fig. 3J). Mice injected with tumor cells alone were used as controls. Highlighting the importance of local production of DKK1, mice co-injected with *Dkk1*-deficient CAFs showed smaller tumor size compared to mice co-injected with *Dkk1*-sufficient CAFs (Fig. 3K).

Because DKK1 is also expressed by cancer epithelial cells in TNBC patient samples, next we assessed the relative importance of CAF-derived DKK1 versus tumor cell-derived DKK1. We used the MDA-MB-231 breast cancer cell line, which expresses endogenous DKK1[45]. Similarly to the TNBC TMAs, we observed a strong DKK1 staining in the nucleus of MDA-MB-231 cells (Fig. 3L). Next, we co-injected MDA-MB-231 cells with either *Dkk1*-sufficient CAFs (isolated from tumors in αSMA-*Dkk1*WT-tdT mice) or *Dkk1*-deficient CAFs (isolated from tumors in αSMA-*Dkk1*cKO-tdT mice) at 1:1 ratio into the MFP of Nude recipient mice. Strikingly, we found a significant decrease in tumor growth in mice co-injected with *Dkk1* null CAFs compared to those co-injected with *Dkk1*-sufficient CAFs (Fig. 3M). These findings highlight the importance of local DKK1 production by CAFs regardless of DKK1 expression in the tumor cells.

## DKK1 has immunomodulatory effects

To assess whether DKK1 exerts direct effects on tumor cell proliferation, we cultured the PyMT, 4T1, and E0771 tumor cells in the presence of recombinant DKK1 (rDKK1) and performed an MTT assay. For all three cell lines, rDKK1 did not increase cell density compared to

unstimulated cells at all time points and doses tested (Supplementary Fig. 4A–C), nor it induced any significant changes in cell cycle and survival (Supplementary Fig. 4D, E).

To better understand how DKK1 promotes tumor progression in vivo, we performed bulk RNA sequencing using GFP-H2B-mApple-Thy1.1⁺ PyMT-BO1 cells isolated from orthotopic tumors in WT mice receiving IgG or mDKN01, after exclusion of Ter119⁺ erythrocytes and CD45⁺ immune cells (Fig. 4A, Supplementary Fig. 4F). Only 134 genes were differentially expressed (DEGs, *p* < 0.05 and |fold change| >2) between the two groups (Fig. 4B). KEGG pathway enrichment analysis confirmed no differences in pathways related to cell viability or cell cycle but rather showed changes in pathways related to immune responses (Fig. 4C). Gene set enrichment analysis (GSEA) further showed hallmarks of anti-tumor immune responses being upregulated in the mDKN01-treated tumors compared to IgG, including interferon-gamma response, interferon-alpha response, TNFα signaling via NFκB, and IL-2/STAT5 signaling (Fig. 4D). These results suggest that expression of DKK1 at tumor site might contribute to create an immune suppressive environment, rather than directly affecting tumor growth.

## Local production of DKK1 at tumor site affects tumor immune infiltration

To determine if DKK1 modulates the immune landscape of the TME, we profiled PyMT tumor-infiltrating immune populations from IgG or mDKN01-treated mice. We found a significant increase in the number of CD45⁺ cells per gram of tumor following mDKN01 administration, with CD4⁺ and CD8⁺ T cells, F4/80⁺ macrophages, and NK cells being the most increased subsets (Fig. 5A, Supplementary Fig. 5A, B). Similar results were found in orthotopic breast tumors isolated from αSMA-*Dkk1*cKO mice compared to αSMA-*Dkk1*WT mice (Supplementary Fig. 5C)

IHC further indicated the presence of CD45⁺ immune populations in the central regions of the tumor mass in PyMT tumors treated with mDKN01 (Fig. 5B), αSMA-*Dkk1*cKO animals (Fig. 5C) and from WT mice co-injected with tumor cells plus αSMA-*Dkk1*cKO CAFs (Fig. 5D). In contrast, peripheral CD45⁺ cell localization was observed in all control groups. These results suggest that CAF-derived DKK1 can limit the infiltration of immune cells at the tumor site, regardless of the elevated levels of DKK1 in circulation.

## DKK1 targets NK cells to support tumor progression

To identify the immune populations targeted by DKK1, we injected PyMT cells into the MFP of female NSG mice, which lack T, B, and NK cells, and administered IgG or mDKN01. Strikingly, the anti-tumor effects of mDKN01 were fully abrogated in this mouse model (Fig. 5E). Next, we adopted a selective immune cell depletion approach. Mice were either depleted of T or NK cells, or treated with vehicle control and randomized to receive IgG or mDKN01. As expected, the depletion of T cells only slightly increased tumor burden compared to IgG controls (Fig. 5F, Supplementary Fig. 5D), confirming the limited involvement of T cells in the PyMT tumor model[46]. Furthermore, mDKN01 anti-tumor effects were not affected by T cell depletion. In contrast, depletion of NK cells significantly increased tumor burden compared

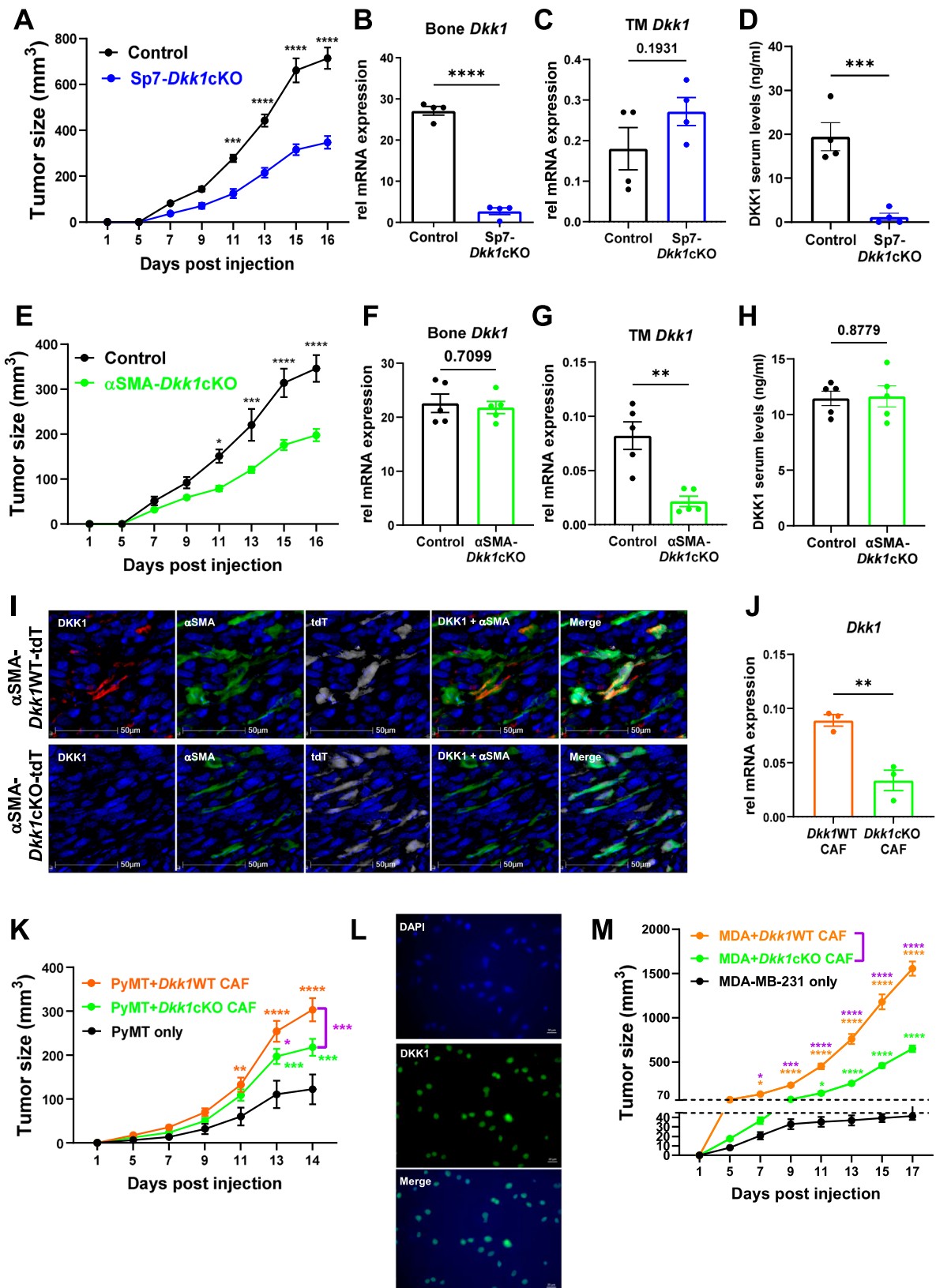

to control mice and completely abrogated the anti-tumor effects of mDKN01 (Fig. 5G, Supplementary Fig. 5E). Similar findings were observed in mice lacking perforin ($Prf1^{-/-}$), a key effector molecule utilized by T and NK cells to form pores enabling the transfer of cytotoxic granzymes to induce the specific killing of target cells (Fig. 5H). Since T cell depletion did not alter the anti-tumor effects of

mDKN01, findings in $Prf1^{-/-}$ mice suggest that DKK1 supports tumor progression through suppression of NK cell function.

## DKK1 directly suppresses NK cell cytotoxicity

NK cells must reside near their target cells to exert cytotoxic effects. We observed that NK cells exposed to rDKK1 (200 ng/ml) for 3 h were

**Fig. 3 | Bone and CAF-derived DKK1 contribute to systemic and local increases in DKK1 levels during tumor progression. A** Tumor growth was determined by caliper measurements in 6–8 weeks old Sp7-*Dkk1*WT (control) and Sp7-*Dkk1*cKO female mice (*n* = 4 mice/group) inoculated with PyMT in the MFP. **B, C** qRT-PCR for *Dkk1* expression in bone and primary tumor (TM). **D** DKK1 serum levels measured by ELISA. **E** Tumor growth was determined by caliper measurements in 10–12 weeks old control and αSMA-*Dkk1*cKO (*n* = 5 mice/group). **F, G** qRT-PCR for *Dkk1* expression in bone and primary tumors (TM). **H** DKK1 serum levels measured by ELISA. **I** Multiplex immunohistochemistry (IHC) of orthotopic PyMT tumors in αSMA-*Dkk1*WT (top, *n* = 6) or αSMA-*Dkk1*cKO mice (bottom, *n* = 6) stained for DKK1 (red), αSMA (green), tdTomato (white) and hematoxylin (blue). **J** qRT-PCR for *Dkk1* expression in sorted tdT⁺ CAFs from primary tumors in αSMA-*Dkk1*WT-tdT and αSMA-*Dkk1*cKO-tdT mice (*n* = 3/group). **K** Tumor growth was determined by

caliper measurements in WT mice co-injected with $10^5$ PyMT cells and $10^5$ tdT⁺ CAFs isolated from αSMA-*Dkk1*WT-tdT or αSMA-*Dkk1*cKO-tdT (*n* = 8/group). Mice injected with tumor cells alone (*n* = 4) were used as control. **L** Representative immunofluorescence images of MDA-MB-231 cells stained for DKK1 (green) and DAPI (blue) (*n* = 3). **M** Tumor growth was determined by caliper measurements in Nude mice co-injected with $10^5$ MDA-MB-231 cells and $10^5$ tdT⁺ CAFs isolated from αSMA-*Dkk1*WT-tdT or αSMA-*Dkk1*cKO-tdT (*n* = 6/group). Mice injected with tumor cells alone (*n* = 6) were used as control. Results are shown as mean ± SEM. An unpaired *t*-test with a two-tailed *P*-value (**B**–**D**, **F**–**H**, **J**), two-way ANOVA followed by Bonferroni multiple-comparison test (**A**, **E**, **K**, **M**) were used to determine significance. *$P < 0.05$, **$P < 0.01$, ***$P < 0.001$, ****$P < 0.0001$. Source data are provided as a Source Data file.

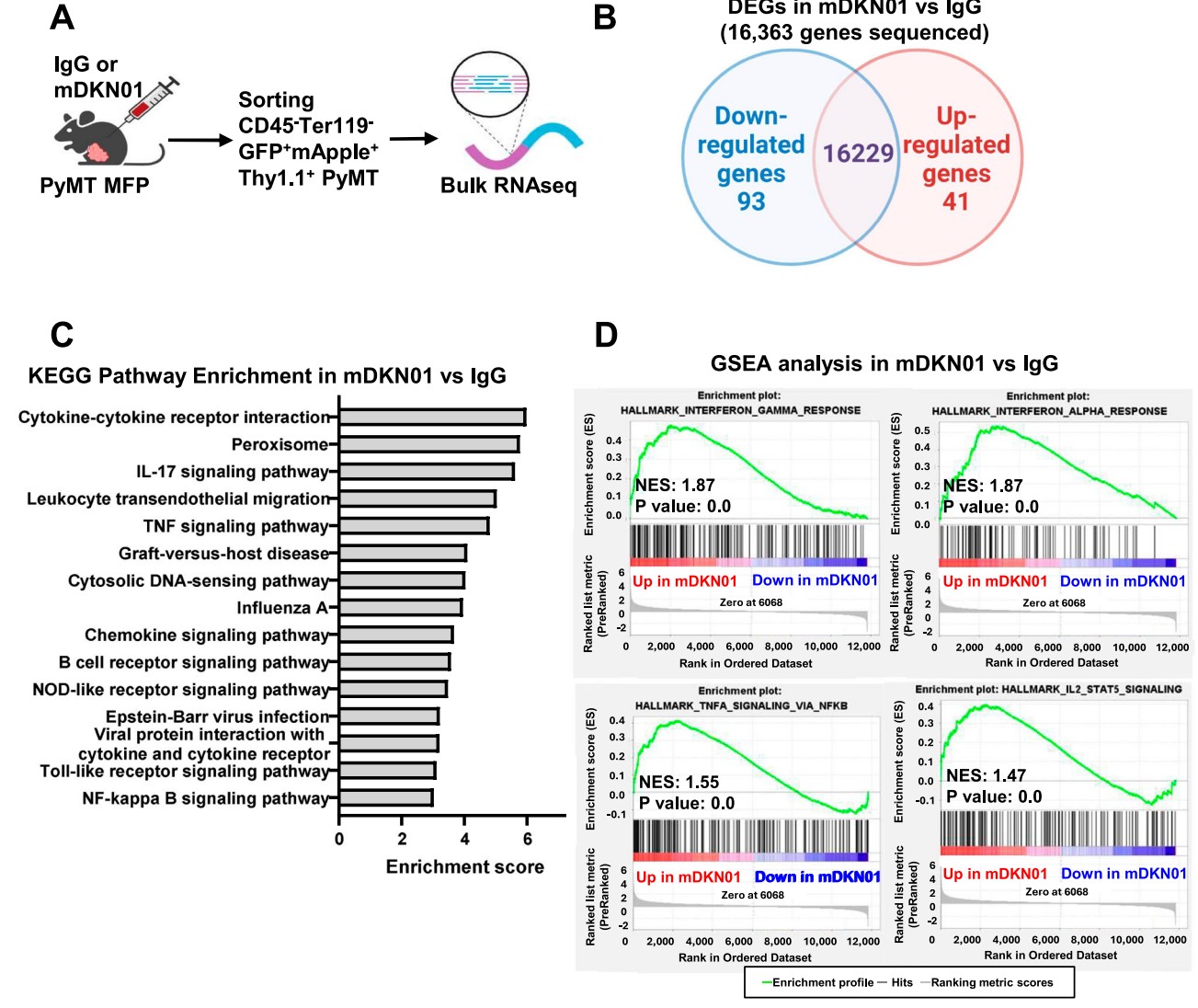

**Fig. 4 | DKK1 has immunomodulatory effects. A** Schematic representation of PyMT tumor cell isolation from IgG or mDKN01-treated mice (*n* = 3/group) and analysis of transcriptome via bulk RNA sequencing. **B** Venn diagram depicting uniquely and commonly expressed genes in PyMT cells isolated from orthotopic tumors injected into WT mice treated with IgG or mDKN01. **C** KEGG pathway

enrichment analysis on differentially expressed genes (DEGs, *p* < 0.05, |fold change| > 2). **D** GSEA analysis of hallmarks upregulated in PyMT tumor cells isolated from mDKN01-treated mice compared to IgG-treated mice. Normalized enrichment score (NES) and nominal *P*-value were calculated as previously described[59].

rarely found in close contact with mCherry⁺ PyMT tumor cells, as shown by high magnification confocal microscopy of F-actin-stained cells (Fig. 6A rDKK1). In contrast, NK cells not exposed to rDKK1 were often in direct association with the tumor cells (Fig. 6A control). Based on these observations, we aimed to determine whether DKK1 directly

affects NK cell cytotoxicity, and quantified NK cell-mediated killing of the cell trace violet (CTV)-labeled PyMT tumor cells by assessing the expression of 7-AAD via flow cytometry (Fig. 6B, Supplementary Fig. 6A). rDKK1 significantly suppressed NK cell-mediated killing at all tested effector (NK cells) to target (PyMT) ratios (Fig. 6C). Similar

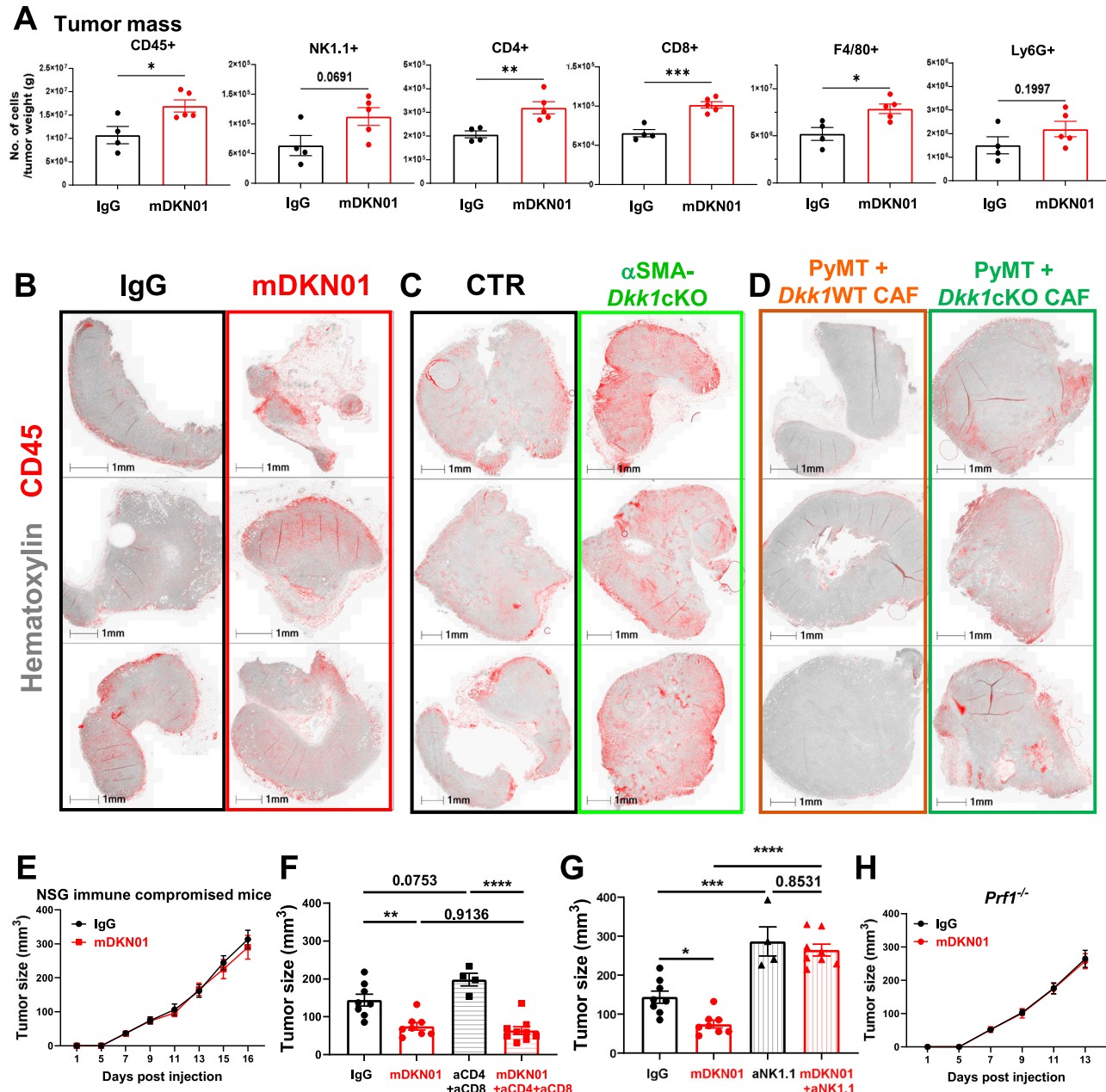

**Fig. 5 | DKK1 supports tumor progression by targeting NK cells. A** Tumor-infiltrating CD45⁺ immune cells, NK cells, T cells, and myeloid subsets per gram of PyMT tumor mass from WT mice treated with IgG (*n* = 4) or mDKN01 (*n* = 5). **B−D** Deconvoluted IHC images from orthotopic PyMT tumors stained for CD45 (red) and hematoxylin (gray) isolated from WT mice treated with IgG or mDKN01 (**B**, *n* = 5/group), αSMA-*Dkk1*WT and αSMA-*Dkk1*cKO mice (**C**, *n* = 5/group), and mice co-injected with tumor cells and tdT⁺ CAFs from tumors in αSMA-*Dkk1*WT-tdT or αSMA-*Dkk1*cKO-tdT mice (**D**, *n* = 8/group). **E** Tumor growth by caliper measurements in 6–8 weeks old NSG immune-compromised mice (*n* = 6 mice/group) inoculated with PyMT into the MFP. **F, G** PyMT orthotopic growth determined by

caliper measurements in 6−8 weeks WT mice treated with IgG (*n* = 8) or mDKN01 (10 mg/kg, *n* = 8) every other day along with anti-CD4 and anti-CD8 (**F**, *n* = 4 and *n* = 9) or anti-NK1.1 (**G**, *n* = 4 and *n* = 8). **H** Tumor growth was determined by caliper measurements in *Prf1⁻/⁻* mice inoculated with PyMT into the MFP and treated i.p. with mDKN01 (10 mg/kg) or control IgG antibody every other day (*n* = 4 mice/group). Results are shown as mean ± SEM. An unpaired *t*-test with a two-tailed *P*-value (**A**), two-way ANOVA followed by Bonferroni multiple-comparison test (**E**, **H**), ordinary one-way ANOVA followed by Dunnett's multiple-comparison test (**F**, **G**) were used to determine significance. *\*P* < 0.05, *\*\*P* < 0.01, *\*\*\*P* < 0.001, *\*\*\*\*P* < 0.0001. Source data are provided as a Source Data file.

results were observed using a 48 h IncuCyte Live Cell assay with a 2:1 effector (NK cells) to target (H2B-mApple-Thy1.1⁺ PyMT-BO1 cells) ratio (Supplementary Fig. 6B).

To investigate whether CAF and bone-derived DKK1 exert similar NK suppressive effects, we co-cultured different numbers of CAFs or osteoblast precursors (pre-OBs), as a source of bone-derived DKK1, together with NK cells and CTV-labeled PyMT tumor cells (2:1 effector to target ratio) and assessed their ability to kill the tumor cells with or

without mDKN01. TdT⁺ CAFs, isolated from orthotopic tumors in αSMA-*Dkk1*WT-tdT mice (Fig. 6D), reduced the NK-mediated tumor killing, and DKK1 neutralization restored NK cell functionality (Fig. 6E). Similarly, pre-OBs isolated from the bone marrow of tumor-bearing mice reduced the NK-mediated killing of PyMT cells (Fig. 6F, G). These inhibitory effects were restored by mDKN01. Collectively, these results demonstrate that both bone-derived and CAF-derived DKK1 significantly suppress NK cell tumoricidal activities.

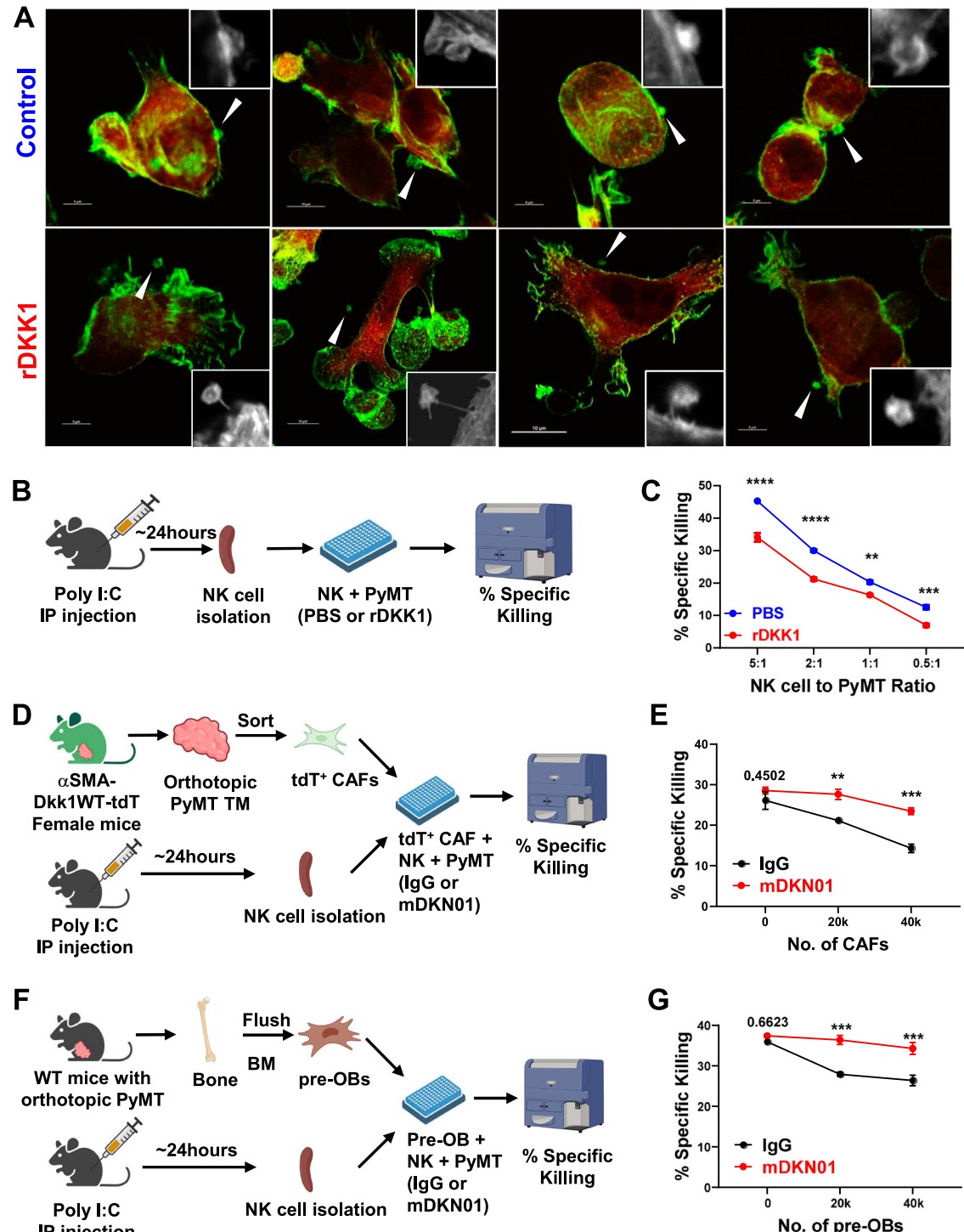

**Fig. 6 | DKK1 suppresses NK cell cytotoxicity. A** Representative immuno-fluorescence images of mCherry⁺ PyMT cells (red) cultured with murine NK cells in the presence of PBS ($n = 11$) or rDKK1 ($n = 9$) for 3 h prior to fixation and staining for F-actin (green). Small, mCherry⁻ cells visualized by arrows or by image contrast in insets represent NK cells. Images in the insets are enlarged 1.5 times. **B** Schematic representation of NK cell isolation from the spleen of Poly I:C treated WT mice ($n = 4$) and incubation with PyMT target cells in the presence of PBS or rDKK1. **C** Analysis of percent specific killing measured by 7-AAD⁺ PyMT target cells after 4 h incubation with NK cells. ($n$ of wells per condition = 3). **D−G** Schematic repre-sentation of CAF isolation from orthotopic PyMT tumors in αSMA-*DKK1*WT-tdT mice (**D**, $n = 5$), osteoblast precursor (pre-OB) isolation from the bone marrow of tumor-bearing WT mice (**F**, $n = 3$) and isolation of NK cells from the spleen of Poly I:C injected mice ($n = 4$). NK cells and increasing numbers of CAFs/pre-OBs were incubated with PyMT target cells in the presence of mDKN01 or IgG (50 mg/ml) and percent specific killing measured by 7-AAD⁺ PyMT target cells after 4 h incubation with NK cells (2:1 NK cell to PyMT ratio) (**E, G** $n$ of wells per condition = 3). Results are shown as mean ± SEM. Experiments in (**C, E, G**) were performed in triplicates. Two-way ANOVA followed by Bonferroni multiple-comparison test was used to determine significance (**C, E, G**). **$P < 0.01$, ***$P < 0.001$, ****$P < 0.0001$. Source data are provided as a Source Data file.

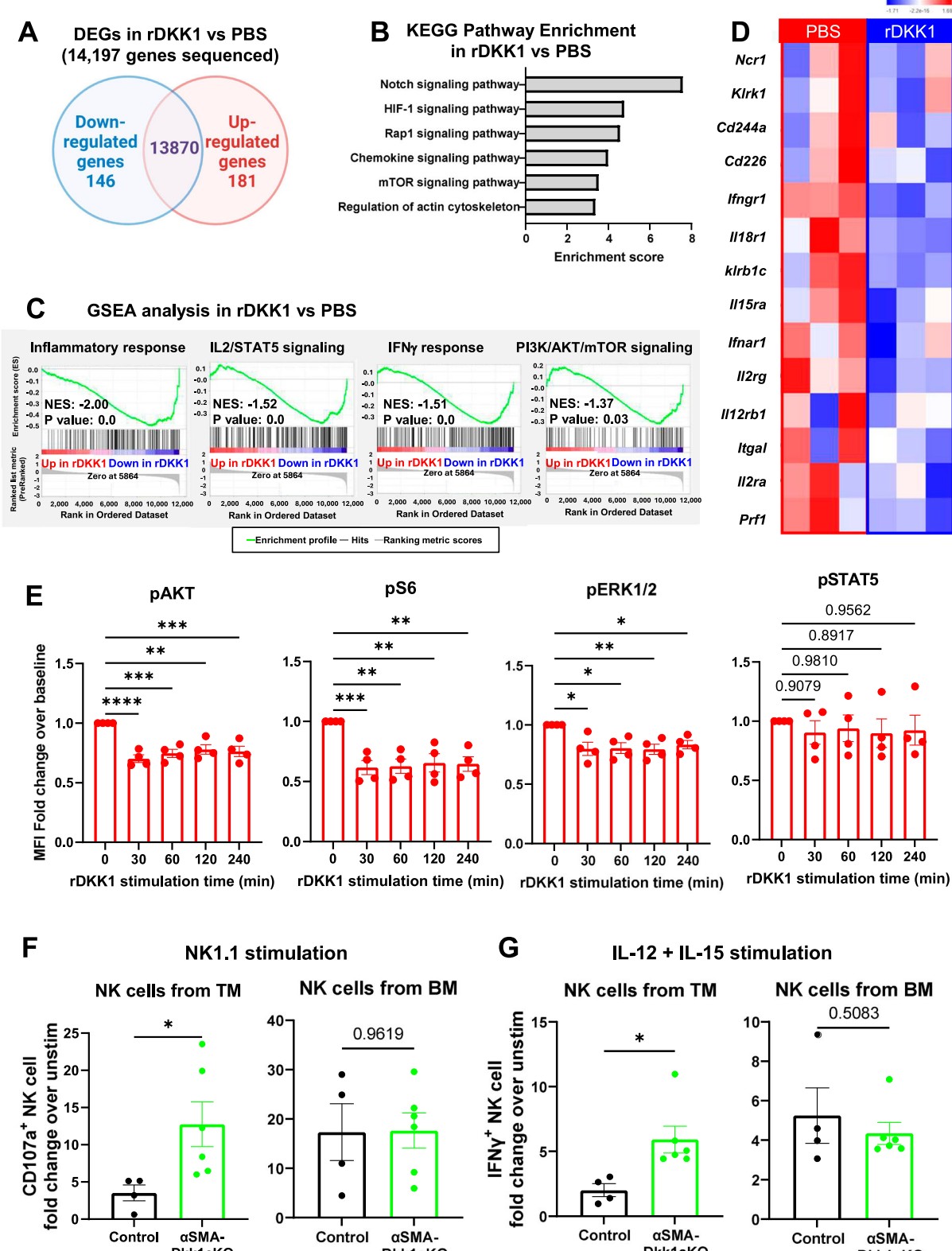

## DKK1 downregulates PI3K/AKT/mTOR and MAPK/ERK pathways in NK cells

To determine how DKK1 impacts NK cells, we performed bulk RNA sequencing of NK cells sorted from the spleen of poly I:C treated mice using the CD45$^+$CD3$^-$NK1.1$^+$ markers and stimulated ex vivo with rDKK1 for 4 h. Out of 14,197 genes detected, 327 genes were differentially

expressed in the rDKK1-stimulated versus unstimulated NK cells (Fig. 7A). KEGG pathway analysis showed enrichment in signaling pathways related to NK cell development (Notch pathway) and function (HIF-1, Rap1, mTOR, and pathways involved in actin cytoskeleton regulation) (Fig. 7B). Furthermore, GSEA analysis showed reductions in genes related to anti-tumor immune responses including the IL-2/

**Fig. 7 | DKK1 downregulates PI3K/AKT/mTOR and MAPK/ERK signaling pathways in NK cells. A** Venn diagram indicating numbers of uniquely and commonly expressed genes in NK cells isolated spleen of Poly I:C treated WT mice ($n = 3$) and stimulated with rDKK1 (200 ng/ml) or PBS as a control ($n = 3$/group). **B** KEGG pathway enrichment analysis showing differentially expressed genes (DEGs, $p < 0.05$, |fold change | >2). **C** GSEA analysis showing hallmarks downregulated in rDKK1 stimulated NK cells. Normalized enrichment score (NES) and nominal *P*-value were calculated as previously described[59]. **D** Heatmap of genes related to NK cell cytotoxicity in PBS or rDKK1 stimulated NK cells. **E** Fold changes from baseline of mean fluorescence intensity (MFI) measurements of phosphorylated AKT, S6, ERK1/2, and STAT5 in NK1.1$^+$ NK cells from the spleens of $Rag1^{-/-}$ mice ($n = 4$) following stimulation with rDKK1 (200 ng/ml) for indicated times. **F, G** Fold changes in the percentage of NK cells expressing CD107a (**F**) or IFNγ (**G**) following ex vivo stimulation with anti-NK1.1 antibody (**F**) or a cytokine cocktail of IL-12 + IL-15 compared to unstimulated cells were evaluated in the PyMT tumor mass and bone marrow of αSMA-*Dkk1*WT ($n = 4$) or αSMA-*Dkk1*cKO mice ($n = 6$). Results are shown as mean ± SEM. Ordinary one-way ANOVA followed by Dunnett's multiple-comparison test (**E**), and an unpaired *t*-test with a two-tailed *P*-value (**F**, **G**) were used to determine significance. *$P < 0.05$, **$P < 0.01$, ***$P < 0.001$, ****$P < 0.0001$. Source data are provided as a Source Data file.

STAT5, the interferon-gamma and the PI3K/AKT/mTOR signaling (Fig. 7C). Accordingly, the rDKK1-exposed NK cells showed decreased gene expression of cytokine receptors *Ifngr1, Ifnar1, Il2r, Il12rb1, Il15ra* and *Il18r1*, adhesion molecules involved in the maintenance of the immunological synapse *Itgal, Cd244a*, and *Cd226* and activating receptors *Ncr1, Klrk1, Klrb1c*, and *Cd226* (Fig. 7D). We also observed that the expression of *Prf1* was decreased in the NK cells exposed to rDKK1.

Given the importance of PI3K[47], MAPK[48], and JAK/STAT[49] in regulating NK cell activation, we investigated whether these signaling pathways were directly modulated by DKK1. We analyzed the phosphorylation of AKT (downstream of PI3K), ribosomal S6 protein (downstream of mTOR), ERK1/2 (downstream of MAPK), and STAT5 in NK cells from the spleens of $Rag1^{-/-}$ mice, which lack T and B cells, thereby making the NK cell gating and intracellular FACS analysis more efficient. Splenocytes were stimulated ex vivo with rDKK1 and NK cells were gated based on NK1.1 expression. We observed reduced phosphorylation of AKT, ribosomal S6 protein, and ERK1/2 in response to rDKK1, but no changes in STAT5 phosphorylation (Fig. 7E, Supplementary Fig. 6C). Interestingly, we did not observe any activation of AKT/ERK/S6 pathways when NK cells were stimulated with the canonical Wnt ligand rWnt3a (Supplementary Fig. 6D). However, when NK cells were stimulated with rWnt3a plus rDKK1 we observed a significant reduction in AKT/ERK/S6 phosphorylation, similar to rDKK1 stimulation alone (Supplementary Fig. 6E). These results suggest that rDKK1 modulates NK cell activation independent of canonical Wnt signaling.

To validate these findings in vivo and assess the local effects of DKK1 on NK cell activation, we examined NK cell activation in orthotopic tumors or the bone marrow from αSMA-*Dkk1*WT and αSMA-*Dkk1*cKO mice. The single-cell suspensions of the tumor mass or bone marrow were stimulated for 4 h with the NK1.1 antibody or a cytokine cocktail containing IL-12 and IL-15, followed by analysis of NK cell degranulation via CD107a surface expression and NK cell activation by IFNγ staining. We found a significant increase in CD107a$^+$ and IFNγ$^+$ NK cells in the tumor mass of αSMA-*Dkk1*cKO compared to αSMA-*Dkk1*WT, but no differences in the bone marrow, where DKK1 levels are similar between αSMA-*Dkk1*cKO and αSMA-*Dkk1*WT mice (Fig. 7F, G, Supplementary Fig. 6F). These results demonstrate the local effects of CAF-derived DKK1 in suppressing NK cell activation in the TME.

### DKK1 directly suppresses human NK cell functions

To investigate whether DKK1 suppresses human NK cells (hNK), we isolated hNK cells from healthy donor PBMCs (Fig. 8A) and found that recombinant human DKK1 (rhDKK1) significantly decreased hNK cell cytotoxicity against the MDA-MB-231 TNBC cell line (Fig. 8B), the T47D ER$^+$ breast cancer cell line (Fig. 8C) and the NK-sensitive K562 cell line (Fig. 8D). Furthermore, rhDKK1 led to a significant decrease in hNK cell activating receptors NKG2D, NKp30, and NKp46 (Fig. 8E, Supplementary Fig. 7A). Interestingly, we did not observe changes in the expression of NK cell activating and/or inhibitory ligands on MDA-MD-231 breast cancer cells following rhDKK1 stimulation (Supplementary Fig. 7B–D). These results were also confirmed using a microarray

dataset from human breast cancer samples (GSE3744[33], Supplementary Fig. 7E–G). Collectively, these findings suggest that in human breast cancer, DKK1 directly suppresses NK cells, rather than modulating the tumor cells.

### DKK1 levels correlate with metastatic progression and reduced cytotoxic NK cells in breast cancer patients

To determine whether DKK1 levels correlate with tumor progression and immune suppression in breast cancer patients, we analyzed DKK1 plasma levels and the activation status of NK cells in the blood of 15 patients with stage IV, HER2$^-$, ER$^+$ breast cancer and skeletal disease, at time of diagnosis and after 15–18 months of standard-of-care endocrine therapy plus Denosumab as antiresorptive therapy. 7 patients were classified as stable as they had no radiographic evidence of skeletal metastatic progression during the study period. 8 patients were classified as progressive as they demonstrated radiographic evidence of new or progressive skeletal lesions (Fig. 8F, Table 1). Although DKK1 levels in circulation were not significantly different in the stable and progressive patients at baseline, patients with progressive skeletal metastases demonstrated a significant increase in DKK1 compared to the time of diagnosis (Fig. 8G).

To assess whether patients with increased DKK1 had reduced and/or dysfunctional NK cells, we measured the number of CD3$^-$CD56$^+$ NK cells in circulation and found no changes between the stable and progressive patients (Supplementary Fig. 8A, B). However, the percentage of cytotoxic CD16$^+$CD56$^{dim}$ NK cells[50] and expression of perforin (PRF1) and granzyme B (GZMB) were reduced in patients with progressive skeletal metastases (Fig. 8H) compared to baseline.

## Discussion

In this study, we demonstrate that high levels of DKK1 in breast cancer create an immune suppressive microenvironment through direct inhibition of NK cell effector functions, and targeting DKK1 has significant anti-tumor effects in the primary and metastatic settings. Moreover, we report a previously unappreciated role for DKK1 in inducing systemic and local immune suppression. Bone cells contribute to the elevated levels of DKK1 in circulation, while CAFs contribute to DKK1 production in the TME and do not affect DKK1 systemically. Intriguingly, deletion of either source of DKK1 results in a significant anti-tumor effect. Our findings indicate the importance of monitoring both systemic and tumor DKK1 levels in breast cancer patients and warrant the exploration of neutralizing DKK1 to achieve an efficient therapeutic response. These results position DKK1 and its inhibitory effects on NK cells as a barrier to immunotherapies and an important driver of breast cancer progression.

Breast cancer is poorly immunogenic, which accounts for the limited response to immune-based anti-tumor therapies. Local production of DKK1 at the tumor site limits tumor immune infiltration, and DKK1 neutralization or its deletion in CAFs reverses these effects. Differently from the B16F10 melanoma model dependent on T cells[23], T cell depletion in our breast cancer models does not limit the therapeutic efficacy of mDKN01. These results are in line with the inconclusive clinical trials in ER$^+$ breast cancer showing limited effects of T

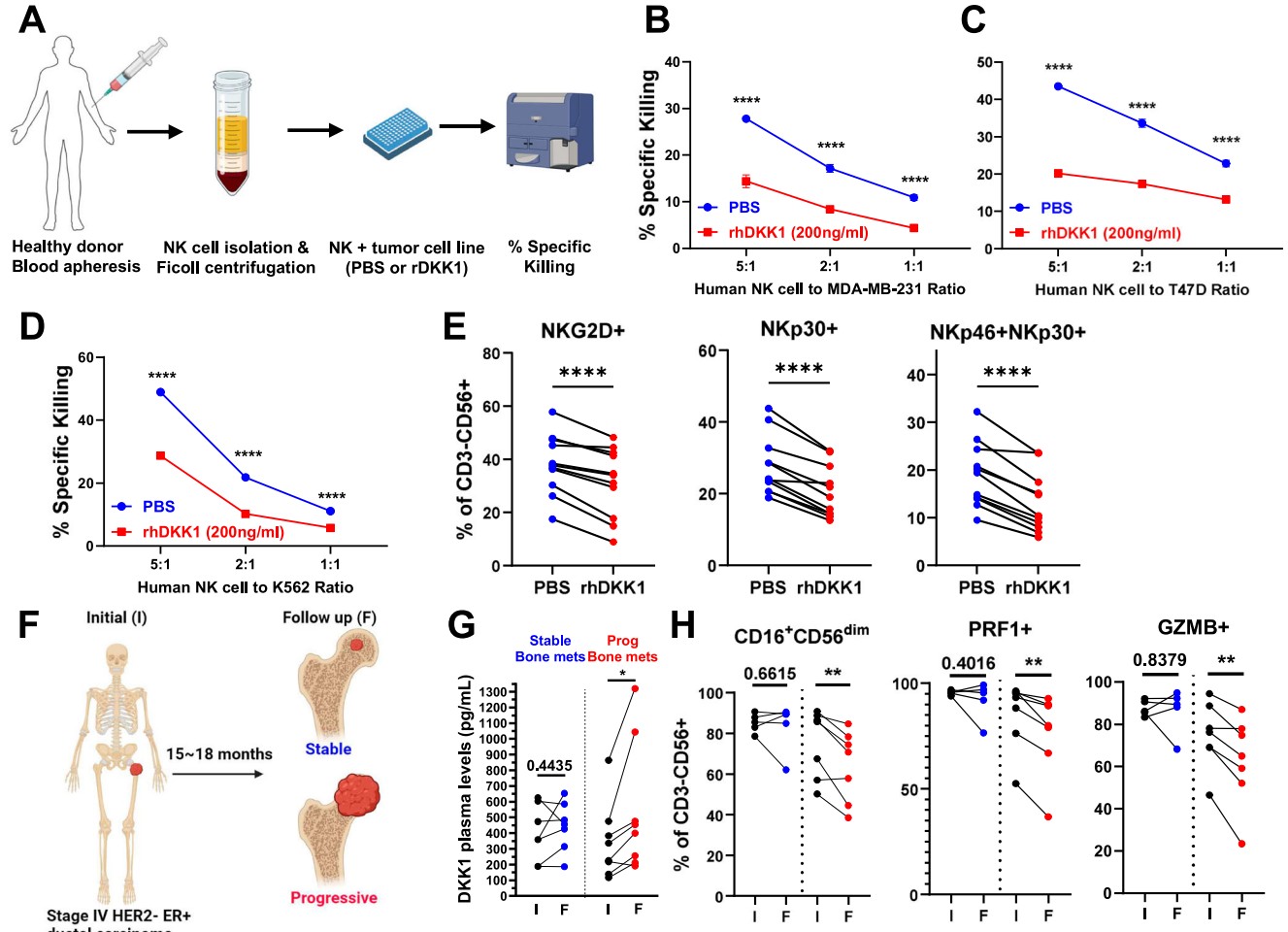

**Fig. 8 | DKK1 levels correlate with reduced therapeutic responses and cytotoxic NK cells. A** Schematic representation of NK cell isolation from healthy donor human PBMCs and incubation with target cells in the presence of PBS or rhDKK1 for 4 h. **B–D** Analysis of percent specific killing measured by 7-AAD+ MDA-MB-231 (**B**, $n = 7$ NK donors), T47D (**C**, $n = 3$ NK donors), and K562 (**D**, $n = 3$ NK donors) target tumor cells after 4 h incubation with human NK cells in the presence of rhDKK1 or PBS as a control. **E** FACS analysis of NK cell activating receptors on human NK cells from different donors ($n = 11$) stimulated with rhDKK1 for 24 h or PBS. **F** Schematic representation of blood sample collection of advanced breast

cancer patients at the time of diagnosis of bone metastases and at 15–18 months follow-up visit receiving standard-of-care and antiresorptive treatments. **G, H** DKK1 plasma levels, and percent of NK cell subsets in patients with regressive/stable (blue, $n = 7$) versus progressive bone metastases (red, $n = 8$) from initial diagnosis (abbreviated as I) and follow-up visits (abbreviated as F). Results are shown as mean ± SEM (**B–D**). Two-way ANOVA followed by Bonferroni multiple-comparison test (**B–D**), and a paired $t$-test with a two-tailed $P$-value (**E, G, H**) were used to determine significance. *$P < 0.05$, **$P < 0.01$, ****$P < 0.0001$. Source data are provided as a Source Data file.

cell-based therapies[5]. Instead, we identify NK cells as important modulators of the anti-tumor effects of mDKN01. Our findings are in line with the observation that the intratumoral abundance of NK cells correlates with increased pCR rate in breast cancer patients undergoing neoadjuvant chemotherapy[10] and findings in the B16F0 melanoma model dependent on NK cells[24].

Direct effects of DKK1 on NK cells have never been reported. We demonstrate that DKK1 suppresses NK cell activation and cytotoxicity. DKK1 directly reduces the phosphorylation of AKT, S6, and ERK1/2, which are key modulators of NK cell activation. These observations are consistent with the transcriptional downregulation of IFNγ and PI3K/AKT/mTOR pathways in NK cells exposed to DKK1. Subsequently, NK cells isolated from a DKK1-deficient TME show improved responses to IL-12, IL-15, and NK1.1 stimulation. The lower levels of perforin, adhesion molecules, and pathways related to cytoskeletal reorganization, together with reduced surface expression of CD107a following NK1.1 engagement, further suggest defects in the formation of immunological synapses with their target cells. In support of this hypothesis, NK cells are not found in close contact

with the target tumor cells when exposed to DKK1. We acknowledge that there are additional potential mechanisms whereby DKK1 may indirectly impact NK cell responses. A prior study identified that DKK1 could induce downregulation of NKG2D and DNAM-1 ligands in a metastatic latency breast cancer model[51]. In addition, there was downregulation of death receptors (FAS and TRAILR), which may protect against this mode of NK cell killing. This idea is further supported by the evaluation of NKG2D-CAR T cells in gastric cancer, whereby DKK1 suppressed NKG2D ligands, reducing NKG2D-based recognition[52]. Other studies have identified that DKK1-induced macrophages and immature myeloid populations become immunosuppressive, thereby inhibiting effector CD8+ T cells and NK cells in gastric cancer[23,26]. In future studies, these mechanisms will be investigated, as well more general impact on the tumor microenvironment. Importantly, DKK1 levels should be monitored in future NK cell-directed clinical trials due to its NK suppressive effects that were not previously appreciated.

Our findings demonstrate the importance of monitoring various sources of DKK1 during tumor progression. *DKK1* is expressed in

**Table 1 | Characteristics of breast cancer patients including time of sample collection (initial diagnosis of metastatic dissemination to bone and at 15–18-month follow-up visit), skeletal and visceral metastasis (met) status, and treatments received at the indicated time. All patients received subcutaneous (SQ) Denosumab 120 mg every 4–12 weeks as an antiresorptive therapy at the time of diagnosis of skeletal metastases**

| Bone met status | Timeline | Bone metastasis | Visceral metastasis | Treatments |
|---|---|---|---|---|
| Stable | Initial<br>Follow up | Sternum, Sacrum, Ilium, Lumbar Spine<br>No changes | Nodal disease<br>Increase R axillary nodes | Letrozole |
| | Initial<br>Follow up | Spine, Sternum, Clavicle, Pelvis<br>No Changes | None<br>None | Anastrozole |
| | Initial<br>Follow up | Spine<br>No changes | Lung<br>No changes | 1. Fulvestrant<br>2. Zoladex |
| | Initial<br>Follow up | Sternum, Ilium<br>No changes | None<br>None | Anastrozole |
| | Initial<br>Follow up | Sternum<br>No changes | Lung, Liver, Lymph node<br>Progressive Liver mets | 1. Everolimus + Exemestane<br>2. Capecitabine |
| | Initial<br>Follow up | Glenoid<br>No changes | None<br>None | Fulvestrant |
| | Initial<br>Follow up | Sternum, Pelvis<br>No changes | None<br>None | Letrozole |
| Progressive | Initial<br>Follow up | Spine, Sternum, Clavicle, Pelvis, Scapula<br>Progression of mets in the axial skeleton | Lymph node, Pleural mass, soft tissue<br>Progessive hepatic mets | 1. Tamoxifen<br>2. Letrozole + Palbociclib |
| | Initial<br>Follow up | Spine, Sacrum<br>Increase T4-T6 Spine Mets | Lung<br>Progessive Lung nodules | Letrozole + Palbociclib |
| | Initial<br>Follow up | Spine<br>New humerus met | None<br>None | Anastrozole |
| | Initial<br>Follow up | Ribs, Spine, Sternum<br>Progression of osseous mets | Liver<br>Increase Liver mets | 1. Everolimus + Exemestane<br>2. Tamoxifen |
| | Initial<br>Follow up | Spine, Ribs, Ilium, Sacrum<br>Progessive mets, New Spine, Femur, Ischium mets | None<br>None | Aromasin |
| | Initial<br>Follow up | Vertebra, Pelvis<br>New Humerus met | Lymph node, Liver<br>Progressive nodal met | 1. Fulvestrant<br>2. Letrozole + Palbociclib |
| | Initial<br>Follow up | Diffuse multifocal osseous mets<br>Progression of multifocal osseous mets | None<br>None | Palbociclib + Tamoxifen |
| | Initial<br>Follow up | Diffuse multifocal osseous mets<br>Progression of multifocal osseous mets | Lung, Liver<br>Progression of Lung/Liver mets | Tamoxifen + AKT inhibitor |

cancer epithelial cells in TNBC and HER2+ tumors, in CAFs, and in the bone. Surprisingly, we observe a strong nuclear DKK1 staining in the tumor cells, which is not typical for a secreted protein, although we cannot exclude that DKK1 has been secreted. Nuclear DKK1 has been previously associated with chemoresistance and poor outcomes in colorectal cancer, where DKK1 binds to specific chromatin sites to regulate the expression of genes involved in the detoxification of chemotherapeutic agents[40]. In contrast, a more diffused DKK1 localization appears in CAFs. Co-culture assays demonstrate that DKK1 produced by CAFs reduces NK cell cytotoxicity, a process restored by DKK1 neutralization. Furthermore, the growth of MDA-MB-231 tumor cells, which display nuclear DKK1 localization, is reduced when these cells are co-injected with Dkk1-deficient CAFs compared to Dkk1-sufficient CAFs. These results highlight the importance of CAF-derived DKK1 in supporting tumor progression regardless of DKK1 tumoral expression, although we cannot exclude that tumor cells also contribute to the release of DKK1 in the TME in patients. Interestingly, Dkk1 is also highly expressed in bone and its deletion exerts profound antitumor effects. Like CAFs, DKK1-producing pre-osteoblasts reduce NK cell cytotoxicity, which can be restored by DKK1 neutralization. It is still unclear why bone and CAF-derived DKK1 do not compensate for each other, and that deletion of Dkk1 in one or the other compartment has such profound anti-tumor effects. Ex vivo NK cell reactivation assays clearly demonstrate that deletion of Dkk1 in CAFs improves NK cell responses in the TME but not in the bone marrow. Since bone is the major site for hematopoiesis, we cannot exclude that bone-derived DKK1 might alter NK cell progenitors by reprogramming hematopoiesis, a process in part modulated by DKK1[53].

Bone is a preferred organ for breast cancer metastatic dissemination. DKK1 is increased in breast cancer patients with progressive bone metastases compared to those with stable disease or visceral metastases[15]. While the source of DKK1 in these patients is unknown, the low expression of DKK1 in the ER+ tumors by IHC and scRNAseq suggests that bone might be the primary site for DKK1 production in ER+ breast cancer. The supportive role of DKK1 on metastatic dissemination to bone has been associated with the local activation of osteoclast-mediated bone resorption[29]. However, our clinical findings show that high levels of DKK1 correlate with the progression of bone metastatic disease despite treatment with anti-resorptives. Instead, we find that the percentages of circulating cytotoxic NK cells were reduced in the progressive patients, suggesting that NK cells might also modulate tumor growth in bone.

In sum, our findings position DKK1 as a barrier for efficient anti-tumor immunity in breast cancer through its suppressive effects on NK cell activation and killing efficiency. Monitoring DKK1 levels in circulation and in the TME, and DKK1 neutralization should be considered to improve the therapeutic response of NK-based strategies and/or the limited efficacy of immunotherapies in breast cancer.

## Methods
This study complies with all relevant ethical regulations including protocols approved by the Institutional Animal Care and Use Committee and guidelines set by the Institutional Review Board of Washington University (IRB ID#: 201102244) along with federal and state guidelines.

## Cell culture

Polyoma middle tumor-antigen murine mammary tumor cells (PyMT, C57BL/6), mCherry-conjugated PyMT (PyMT-mCherry), PyMT-derivative PyMT-BO1 conjugated with firefly luciferase (PyMT-BO1-fluc), H2B-mApple and Thy1.1 conjugated PyMT-BO1 (PyMT-BO1-GFP-fluc-H2B-mApple-Thy1.1), E0771 murine mammary tumor cells (E0771-fluc, C57BL/6), 4T1 murine mammary tumor cells (4T1-GFP-fluc, BALB/c), T47D and MDA-MB-231 human breast cancer cells were cultured at 37 °C with 5% $CO_2$ in complete media (DMEM supplemented with 100 µg/ml streptomycin, 100 IU/ml penicillin, and 1 mM sodium pyruvate) containing 10% FBS. K562 cell line was cultured in complete media (RPMI 1640 supplemented with 2 mM L-glutamine, 100 µg/ml streptomycin, 100 IU/ml penicillin, 1× nonessential amino acids, and 1 mM sodium pyruvate) containing 10% FBS. All cell lines tested negative for *Mycoplasma* contamination during routine evaluations. Aliquots for each cell line were used for a maximum of 1 month after the initial thaw.

## Animal models

Because the breast cancer cell lines used in this study were obtained from female mice, we have restricted our analyses to females. Female wild-type (WT) C57BL/6, WT BALB/c, B6(Cg)-*Tyr$^{c-2J}$*/J (albino C57BL/6), B6.129S7-*Rag1$^{tm1Mom}$*/J (*Rag1$^{-/-}$*), Nu/J (Nude), NOD.Cg-*Prkdc$^{scid}$Il2rg$^{tm1Wjl}$*/SzJ (NSG), C57BL/6-*Prf1$^{tm1Sdz}$*/J (*Prf1$^{-/-}$*), Sp7 Cre (Sp7-tTA, tetO-EGFP/Cre) mice were purchased from The Jackson Laboratory. Mice arrived at 4–6 weeks of age and were allowed to recover from shipping stress and acclimatize to the new environment for at least 2 weeks before use in experiments. αSMACreER$^{T2}$ transgenic mice were a generous gift from Dr. Ivo Kalajzic (University of Connecticut Health Center, Farmington, CT[54]). FSP1 Cre mice were a generous gift from Dr. Regis J. O'Keefe (Washington University, St. Louis, MO[55]). *Dkk1* floxed mice were a generous gift from Seppo J. Vainio (University of Oulu, Finland[56]). In experiments using conditional KO mice, Cre$^+$;*Dkk1*WT or Cre$^-$;*Dkk1*$^{fl/fl}$ were used interchangeably as littermate controls. Animals were housed in a pathogen-free animal facility at Washington University (St. Louis, MO) with a 12-h light/12-h dark cycle and 20-23 °C and 40-60% humidity housing conditions. All experiments were performed according to protocols approved by the Institutional Animal Care and Use Committee at Washington University (Protocol ID: 2022-0315).

To establish tumors, PyMT, PyMT-BO1-fluc, PyMT-BO1-GFP-fluc-H2B-mApple-Thy1.1, E0771-fluc, and 4T1-fluc tumor cells were suspended in 1:1 PBS/Matrigel ratio (Corning 354234) and $10^5$ cells injected into the MFP or inoculated subcutaneously in the left flank. For models of metastases, $10^4$ cells suspended in PBS were injected into the right tibia or intracardiacally in female age-matched mice. αSMACreER$^{T2}$ transgenic mice (αSMA-*Dkk1*cKO, αSMA-*Dkk1*WT) were intraperitoneally injected with tamoxifen (Sigma, 100 mg/kg) resuspended in corn oil starting a day prior to tumor inoculation. For MFP and subcutaneous tumors, measurements were performed every other day with a caliper, and volumes were calculated using the following formula: $V = 0.5$ (length [mm] × width [mm]$^2$). The maximal tumor size (<2000 mm$^3$) permitted by the Institutional Animal Care and Use Committee was not exceeded. For intratibial and intracardiac tumor delivery, growth curves were determined by bioluminescence imaging using an IVIS 50 imaging system (PerkinElmer, 1–60 s exposures, binning 4, 8, or 16, FOV 15 cm, f/stop1, open filter). Mice were i.p. injected with D-luciferin (150 mg/kg in PBS; Gold Biotechnology) and imaged 10 min later under isoflurane anesthesia (2% vaporized in $O_2$). Bioluminescence photon flux (photons per second) data were analyzed by region of interest measurements (fixed region of interest over the whole body, or hindlimb) in Living Image 3.2 (Caliper Life Sciences). For i.c. injections, mice with extra pleural intrathoracic tumors were excluded from the analysis.

## In vivo treatments (Neutralizing antibodies)

The monoclonal anti-mouse anti-DKK1 Ab (Leap therapeutics, mDKN01, produced by grafting complementary determining regions of mDKN01 (IgG4/kappa), with minor modifications, onto a murine IgG2a/kappa construct with an FcR incompetent construct D265A substitution to abrogate FcR interactions[57]) was used to neutralize DKK1. Biodistribution analysis of infrared (IR)-dye conjugated mDKN01 or IgG2a antibody was evaluated in mice bearing primary tumors to confirm mDKN01 delivery at tumor site (Supplementary Fig. 9). To assess the anti-tumor effects of DKK1 targeting, treatment consisted of intraperitoneal injections of mDKN01 and control antibodies at a concentration of 10 mg/kg 2–3 times a week.

The monoclonal anti-mouse anti-CD4 Ab (BioXCell, clone GK1.5), anti-mouse anti-CD8a (BioXCell, clone 2.43) were used to deplete T cells and PBS was used as control. Treatment consisted of intraperitoneal injections of anti-CD4 and anti-CD8a into mice 2 days prior to tumor cell implantation using the following regimen: 500µg on the first dose and 250µg for the subsequent doses every 4 days of each antibody.

The monoclonal Ab anti-mouse anti-NK1.1 Ab (Leinco Technologies, Inc., clone PK136), was used to deplete NK cells and PBS was used as control. Treatment consisted of intraperitoneal injections of anti-NK1.1 into mice 2 days prior to tumor cell implantation at a dose of 100µg once a week. Effective depletion of T and NK cells was assessed by flow cytometry of peripheral blood.

## Multiplex IHC

Freshly isolated mouse primary tumors were fixed in 10% neutral-buffered formalin (DiRuscio & Associates, Inc.) for 24 h. Tissues were paraffin-embedded and sectioned 5 µm thick by the histology core of the Washington University Musculoskeletal Research Center.

Human tissue microarrays were prepared by The St. Louis Breast Tissue Registry (funded by The Department of Surgery at Washington University, St. Louis, MO). Data and tissues were obtained in accordance with the guidelines established by the Washington University Institutional Review Board (IRB #201102394) and WAIVER of elements of Consent per 45 CFR 46.116 (d). All patient information was deidentified prior to sharing with investigators.

Tissues were automatically stained via the Bond Rxm (Leica Biosystems) following dewaxing and appropriate epitope retrieval. Immunostaining was chromogenically visualized using the Bond Polymer Refine Detection (#DS9800, Leica Biosystems) or the Bond Polymer Refine Red Detection (#DS9390, Leica Biosystems). Slides were mounted using Xylene-based Cytoseal (Thermo Fisher) or Vectamount (Vector Labs) as appropriate, scanned via a Zeiss AxioScan 7 microscope and IHC analyses performed using the HALO image analysis platform (Indica Labs, Deconvolution v1.1.1, Multiplex IHC v.3.2.3 algorithms). The following antibodies were used: αSMA (Abcam, Cat# ab5694, 1:1500 (Human), 1:200 (Mouse)), COL1a1 (Cell signaling, Cat# 72026, 1:100 (Mouse)), COL14a1 (Cell signaling, Cat# 61964, 1:200 (Human)), DKK1 (Proteintech, Cat# 21112-1-AP, 1:100 (Human), 1:3000 (Mouse)), PDGFRα (Cell signaling, Cat# 5241, 1:200 (Human)), PanCK (Novus, Cat# NBP2-29429, 1:1000 (Human)).

## scRNAseq analysis

For scRNAseq analysis of the human breast cancer dataset, the matrix was downloaded from the European Genome-Phenome Archive (EGA) EGAS00001005173 and analyzed using Seurat version 4. Seurat object generation and further data processing were conducted as described[39]. Non-tumor (negative for KRT8, 18, 14, 17, and EPCAM) non-immune cell populations were re-clustered for in-depth analysis of endothelial cells (positive for PECAM1) and CAF subpopulations (positive for PDGFRB) as shown in ref. 39. The identities of CAF subpopulations were determined based on the expression of CAF

signature genes identified by previous literature[37,38]. Specifically, myCAFs were identified as CAFs positive for *PDGFRA*, *ACTA2*, *THBS2*, and *POSTN*; vascular CAFs (vCAF) were identified as CAFs positive for *MCAM*, *ACTA2*, *DES*, and *NOTCH3*; inflammatory CAFs (iCAF) were identified as CAFs positive for *PDGFRA*, *CXCL12*, *COL14A1*, and *ALDH1A1*, all of which have been illustrated in a previous work[39]. Visualizations of *DKK1* expression were performed based on the adaptively-threshold low-rank approximation (ALRA) assay of the dataset[58].

## Bulk RNAseq analysis

RNA was isolated from the sorted cells using RNeasy Micro Kit (Qiagen). Total RNA integrity was determined using Agilent Bioanalyzer or 4200 Tapestation. Library preparation was performed with 10 ng of total RNA with a Bioanalyzer RIN score greater than 8.0. ds-cDNA was prepared using the SMARTer Ultra Low RNA kit for Illumina Sequencing (Takara-Clontech) per the manufacturer's protocol. cDNA was fragmented using a Covaris E220 sonicator using peak incident power 18, duty factor 20%, and cycles per burst 50 for 120 s. cDNA was blunt-ended, had an A base added to the 3′ ends, and then had Illumina sequencing adapters ligated to the ends. Ligated fragments were then amplified for 12–15 cycles using primers incorporating unique dual index tags. Fragments were sequenced on an Illumina NovaSeq-6000 using paired-end reads extending 150 bases.

Partek Flow software (Partek Inc., St. Louis, MO) was used for data analysis. Briefly, sequenced reads were aligned with STAR 2.7.8a index (Mus musculus – mm10 assembly, Whole genome index). Raw read counts were obtained by quantitating aligned reads using HTSeq with the Ensembl Transcripts release 102 annotation model. Raw read counts were then normalized using counts per million (CPM) and an offset of 0.0001 was added to all normalized read counts. Normalized read counts for each mRNA were statistically modeled using Partek Flow's Gene Specific Analysis (GSA) approach. Differentially expressed genes (DEGs) were then filtered by using P-values less than or equal to 0.05 and fold changes bigger than 2. KEGG pathway enrichment analysis was performed with Partek Flow software (Partek Inc., St. Louis, MO). GSEA analysis was performed as previously described[59,60].

## Multiparametric flow cytometry

Immediately upon sacrifice, single-cell suspensions were prepared from tumors. In brief, tumor tissues were minced, and then digested with 3.0 mg/ml collagenase A (Roche) and 50 U/ml DNase I (Sigma-Aldrich) in serum-free media for 30 min at 37 °C. Cells were filtered through 70 μm nylon strainers (Thermo Fisher Scientific) and washed twice in PBS with 2% FBS. Red blood cells were then removed with red blood cell lysis buffer (Sigma-Aldrich). Cells were washed once, blocked with anti-mouse CD16/CD32 blocker, and stained in PBS with 0.5% BSA, 2 mM EDTA, and 0.01% NaN₃ with the anti-mouse antibodies. The acquisition was performed on a BD LSRFortessa X-20 Cell Analyzer or CYTEK Northern Lights and the dedicated software Diva (BD) or CYTEK SpectroFlo. Data were analyzed with FlowJo 10.9.0 software (Tree Star).

The following antibodies were used: Anti-mouse CD3e-FTIC (Biolegend, Cat# 100306, Clone 17A2, 1:400), CD4-APC (BD Pharmingen, Cat# 561091, Clone RM4-5, 1:200), CD8a-BUV395 (BD Horizon, Cat# 565968, Clone 53-6.7, 1:200), CD11b-BUV395 (BD Biosciences, Cat# 565976, Clone M1/70, 1:400), CD16/32 (Biolegend, Cat# 101302, Clone 93, 1:500), CD45-APC Cy7 (Biolegend, Cat# 103116, Clone 30-F11, 1:400), CD45-BV605 (Biolegend, Cat# 103140, Clone 30-F11, 1:400), CD107a-eFlour660 (Thermo Fisher, Cat# 50-1071-82, Clone 1D4B, 1:200), F4/80-BV711 (Biolegend, Cat# 123147, Clone BM8, 1:100), IFNg-BV750 (Biolegend, Cat# 505865, Clone XMG1.2, 1:50), Ly6C-APC (Biolegend, Cat# 128016, Clone HK1.4, 1:400), Ly6G-BV421 (BD Biosciences, Cat# 562737, Clone 1A8, 1:400), NK1.1-BV711 (Biolegend, Cat# 108745, Clone PK136, 1:200), pAKT-BV421 (BD phosflow, Cat# 562599,

Clone M89-61, 1:20), pERK1/2-PE (BD phosflow, Cat# 612566, Clone 20A, 1:10), pS6-eFlour450 (BD phosflow, Cat# 561457, Clone N7-548, 1:20), pSTAT5-Alexa Fluor647 (BD phosflow, Cat# 612599, Clone 47/Stat5, 1:20), Ter119-BV605 (Biolegend, Cat# 116239, Clone TER119, 1:200), CD90.1- eFlour450 (Thermo Fisher, Cat# 48-0900-82, Clone HIS51, 1:200).

Anti-human CD3-ECD (Immunotech, Cat# A07748, Clone UCHT1, 1:50), CD16-PerCP Cy5.5 (BD Pharmingen, Cat# 560717, Clone 3G8, 1:200), CD45-BV605 (Biolegend, Cat# 304042, Clone 2D1, 1:20), CD56-PE Cy7 (Immunotech, Cat# A51078, Clone N901, 1:100), CD58-PerCP Cy5.5 (Biolegend, Cat# 330914, Clone TS2/9, 1:50), B7-H6-PE (R&D systems, Cat# FAB7144P, Clone 875001, 1:20), GzmB-AF700 (BD Pharmingen, Cat# 560213, Clone GB11, 1:50), HLA-E-APC (Biolegend, Cat# 342606, Clone 3D12, 1:20), NKG2D-APC (Invitrogen, Cat# 17-5878-42, Clone 1D11, 1:20), NKp30-BV785 (Biolegend, Cat# 325229, Clone P30-15, 1:20), NKp46-BV421 (BD Horizon, Cat# 564065, Clone 9E2, 1:20), PRF1-PE (Biolegend, Cat# 308106, Clone dG9, 1:50).

## NK cell killing assays

Cytotoxicity of NK cells was assessed in a standard 4h flow cytometry-based ex vivo killing assay as previously described[61]. $4 \times 10^4$ tumor cells were plated in 96-well plates and different numbers of NK cells were added one hour later. Recombinant mouse DKK1 (Biolegend) or human DKK1 (Biolegend) was added to a final concentration of 200 ng/mL in RPMI 1640 media. For each effector: target (E:T) ratio, Percent Specific Killing was calculated as [% 7-AAD⁺ of CTV⁺ cells](Effector+Target) well − [% 7-AAD⁺ of CTV⁺ cells](Targets only) well.

Specific killing of control and rDKK1-stimulated NK cells was also evaluated using the IncuCyte Live-cell Analysis system (Satorius). A total of $5 \times 10^3$ mApple⁺ PyMT tumor cells were incubated for 2 h in a 96-well plate and imaged prior to the addition of the NK cells at a 2:1 E:T ratio. Real-time images were captured every 4 h and up to 48 h and analyzed using the IncuCyte software. Data are presented as red object counts (mApple⁺ cells).

Osteoblast precursors were generated from bone marrow harvested from tibias and femurs of PyMT tumor-bearing mice by centrifugation and cultured for 7 days on a tissue culture dish in αMEM media at 37 °C. Adherent cells were detached with Trypsin LE Express (Gibco) and plated at increasing numbers 24 h before plating NK cells ($8 \times 10^4$ Cells) and tumor cells ($4 \times 10^4$ Cells) in the presence of IgG or mDKN01 (50 μg/ml). Analysis of NK tumoricidal activity was performed as indicated above.

CAFs were isolated by plating the single-cell suspension of the tumor mass from αSMA-*Dkk1*WT-tdT mice for 30 min on a tissue culture dish at 37 °C, followed by replacing the media to remove immune and tumor cells in suspension. After 20 h of culture, the adherent cells were carefully detached with accutase digestion, stained with anti-CD45, and sorted based on tdT expression and exclusion of CD45⁺ cells. Isolated CAFs were plated at increasing concentrations 24 h before plating NK cells ($8 \times 10^4$ Cells), tumor cells ($4 \times 10^4$ Cells) + IgG or mDKN01 (50μg/ml). Analysis of NK tumoricidal activity was performed as indicated above.

## Analysis of phosphorylated signaling pathways in NK cells

Spleens from *Rag1*⁻/⁻ mice were mechanically disassociated with a 1 cc syringe plunger and passed through a 70 μm cell strainer. Red blood cells were lysed with red blood cell lysing buffer Hybrid Max (Sigma-Aldrich). Splenocytes were washed with PBS and $5 \times 10^5$ cells were plated in 96-well plates in media without IL-15 followed by rDKK1 (200 ng/ml) stimulation at 37 °C for indicated amount of time. After incubation, cells were stained with anti-NK1.1 antibody and fixed with 1% PFA for 10 min at room temperature and permeabilized with ice-cold 100% methanol for 30 min at 4 °C. Cells were washed three times with the FACS buffer, stained with antibodies against pAKT, pS6, pERK1/2, and pSTAT5, followed by acquisition with CYTEK Northern

Lights. Mean fluorescence intensity (MFI) was analyzed from NK1.1$^+$ cells with FlowJo 10.9.0 software (Tree Star).

### Ex vivo NK cell activation

Single-cell suspensions were prepared from the tumor mass and bone marrow from αSMA-*Dkk1*cKO and αSMA-*Dkk1*WT mice. Red blood cells were lysed with red blood cell lysing buffer Hybrid Max (Sigma-Aldrich). Cells were washed with PBS and $1–2 \times 10^6$ cells were plated in media containing 10 ng/mL IL-15 and anti-CD107a flow antibody onto 96-well plates coated with anti-NK1.1 antibody (PK136, 5 μg/mL), or simulated with IL-12 and IL-15 cytokine cocktail (10 ng/ml each). After 1 h at 37 °C, GolgiPlug (BD biosciences) and GolgiStop (BD biosciences) were added for an additional 3 h. Cells were washed with the FACS buffer, and stained with anti-CD45, CD3, CD49b, and NKp46 flow antibodies. IL-12 and IL-15 stimulated cells are then fixed/permeabilized with BD cytofix/cytoperm buffer for 30 min at 4 °C, stained with anti-IFNγ antibody, and acquired on CYTEK Northern Lights. Live NK cells (CD45$^+$CD3$^-$CD49b$^+$NKp46$^+$) were analyzed with FlowJo 10.9.0 software (Tree Star).

### Human NK cell purification and cell culture

Human platelet apheresis donor PBMCs were obtained by Ficoll centrifugation. NK cells were purified using RosetteSep (StemCell Technologies, ≥95% CD3$^-$CD56$^+$). Cells were plated at $3–5 \times 10^6$ cells/mL and cultured in complete RPMI 1640 medium containing 10% human AB serum (Sigma-Aldrich) supplemented with rhIL-15 (1 ng/mL) to support survival, with 50% of the medium being replaced every other day with fresh cytokines.

### Immunofluorescence and Confocal microscopy

MDA-MB-231 tumor cells were plated overnight onto FBS-coated slides, fixed with 4% PFA, and permeabilized with 0.3% Triton X-100. Cells were stained with DKK1 antibody (Proteintech (21112-1-AP)) followed by anti-rabbit-Alexa 488 secondary antibody (Abcam, ab150077, 1:1000) and DAPI (Abcam, ab228549) for an hour and mounted with anti-fade fluorescence mounting medium (Abcam).

PyMT-mCherry tumor cells were plated onto the poly-D-lysine (Millipore sigma) overnight coated slides and incubated for an hour before adding NK cells. NK and tumor cells were cultured together for 3 h prior to fixation with 4% PFA and permeabilization with 0.3% Triton X-100. Cells were stained with an F-actin antibody conjugated with Alexa 488 (Invitrogen, A12379, 1:200) for an hour and mounted with an anti-fade fluorescence mounting medium (Abcam).

Confocal images were generated using a Nikon AX-R Confocal Microscope (100× objective lens, oil immersion, Numerical Aperture: 1.45, Refractive Index: 1.515). The camera setting was set in multi-channel detector mode (FITC and mCherry), Galvano unidirectional scanner with Band scan mode, 4× line averaging, 1.0 μs dwell time. Images were analyzed with NIS-Elements software (Nikon, ver.5.21.00) and ImageJ[62].

### Real-time PCR analysis

RNA was isolated from the cell line or tumor mass using RNeasy Mini Kit (Qiagen). Purified RNA was then reverse transcribed to cDNA using a High Capacity cDNA reverse transcription kit (Applied Biosystems) according to the manufacturer's instructions. The subsequent real-time PCR analysis was performed with SYBR Green PCR Master Mix (Applied Biosystems) and primers specific for murine DKK1 and cyclophilin were used as follows: for *Dkk1*, CTC ATC AAT TCC AAC GCG ATC A (forward), GCC CTC ATA GAG AAC TCC CG (reverse) and for *cyclophilin* AGC ATA CAG GTC CTG GCA TC (forward) and TTC ACC TTC CCA AAG ACC AC (reverse).

For primers specific for human NK cell activating/ inhibitory receptor ligands were used as follows: for *HLA-E*, TTC CGA GTG AAT CTG CGG AC (forward), GTC GTA GGC GAA CTG TTC ATA C (reverse),

*PVR*, GGA CGG CAA GAA TGT GAC CT (forward) GGT CGT GCT CCA ATT ATA GCC T (reverse), *MIC-A*, CTT CAG AGT CAT TGG CAG ACA T (forward), TGT GGT CAC TCG TCC CAA CT (reverse), *PVRL2*, CAC TTG CGA GTT TGC CAC C (forward), GCC ACT GTC GTA GGG TCC T (reverse), *CD58*, AGA GCA TTA CAA CAG CCA TCG (forward), ATC TGT GTC TTG AAT GAC CGC (reverse).

### MTT assay

$5 \times 10^2$ PyMT or 4T1 cells and $2 \times 10^3$ E0771 cells were plated for 24 h before treatment with rDKK1 (50, 100, 200 ng/ml) or tamoxifen (1, 2, 10 μM) for the indicated amount of time. The MTT assay (Invitrogen) and annexin V assay (Thermo Fisher Scientific) were performed per the manufacturer's protocol.

### Human study

All human samples were obtained in accordance with guidelines set by the Institutional Review Board of Washington University (IRB ID#: 201102244) and followed federal and state guidelines. All participants gave written informed consent under the IRB-approved protocol prior to inclusion in the study, including access to archival tumor tissue for research. Samples were deidentified prior to sharing with collaborators. All studies were conducted in compliance with the Declaration of Helsinki. The clinical information of the enrolled patients, including age, timeline of visits, sites of metastatic dissemination and treatment histories, was reviewed and documented by the breast oncologist C.X.M. Patients were all female with a diagnosis of estrogen receptor (ER) positive, HER2 negative breast cancer, stage IV, with bone metastases, prior to first-line systemic therapy for metastatic breast cancer or had prior therapy for metastatic breast cancer but met the following criteria: (i) no prior chemotherapy or immune therapy in the past 2 months, (ii) patients currently stable or progressing on hormonal therapy or hormonal therapy combination or starting hormonal therapy or hormonal therapy combination, and (iii) no limitation on the number of prior hormonal therapy or chemotherapy treatments. Radiologic tumor assessment was required within 1 month prior to or after the collection of the baseline blood sample to serve as the baseline tumor assessment.

Peripheral blood was collected at enrollment and at 15–18 months follow-up visits from 15 patients being treated with Denosumab (anti-RANKL) and on standard-of-care treatment. To isolate peripheral blood mononuclear cells (PBMC), EDTA-treated whole blood was diluted to a volume of 20 mL with PBS, transferred to a 50 mL conical tube, and underlaid with 15 mL of Ficoll (Atlantal Biologicals). Tubes were centrifuged at $400 \times g$ for 30 min. The PBMC fraction was collected at the interface layer and washed three times with 40 mL of PBS. After counting, a minimum of $5 \times 10^6$ PBMCs were frozen in 10% [volume for volume (v/v)] DMSO (Sigma-Aldrich, catalog no. D5879) in FBS and stored in liquid nitrogen for subsequent analysis.

### ELISA

Serum or plasma levels of DKK1 from mice or patients were quantified using ELISA kits specific for DKK1 (Mouse Dkk1 Quantikine ELISA kit or Human Dkk1 Quantikine ELISA kit, R&D systems) as per the manufacturer's protocols.

### Statistical analysis

In vitro experiments include technical and biological triplicates and were performed at least 3 times. In vivo experiments were done with at least 4–9 mice per group (the number of mice used for each experiment is specified in the figure legends) and at least 3 independent experiments were performed. Numerical data are shown as mean ± SEM. *T*-test statistical analysis was used to compare two groups (unpaired *t*-test when comparing murine data sets and paired when comparing human cells from single donors or patients). In calculating two-tailed significance levels for equality of means, equal variances

were assumed for the two populations. In some experiments with multiple groups or time points, analysis of variance (ANOVA), including Dunnett's multiple-comparison test, or Bonferroni's multiple-comparison test, was used. Results were considered significant at $p < 0.05$. All statistical analyses were performed with GraphPad Prism 10.2.1 software for Windows (GraphPad Software).

## Reporting summary

Further information on research design is available in the Nature Portfolio Reporting Summary linked to this article.

## Data availability

The bulk RNA sequencing data generated in this study has been deposited in NCBI's Gene Expression Omnibus[63] and is accessible through GEO Series accession number GSE262733. Previously published data sets are accessible through the GEO Series accession number (GSE3644, GSE8977, GSE176078). The remaining data are available within the Article, Supplementary Information, and in Source data file. Source data are provided with this paper.

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

## Acknowledgements

We thank the Musculoskeletal Histology Core of Musculoskeletal Research Center supported by the National Institutes of Health P30 Grants AR057235 and P30 AR074992, the Washington University Bright Institute, Molecular Imaging Center (NIH P50 CA94056ADD) and Siteman Flow Cytometry Core supported by NCI Cancer Center support grant P30 CA91842 and P30CA091842. This research was supported by grants from National Institutes of Health Grants R01 AR066551 (to R.F.), CA270030 (to R.F.) and CA235096 (to R.F.), grants from Shriners Hospital 85170 and P19-07408 CR (to R.F.), and the Siteman Investment Program, Siteman Cancer Center (Pre-R01 Program to R.F.) and the Foundation for Barnes-Jewish Hospital (3770 and 4642). We thank Leap Therapeutics for providing mDKN01 and M. Kagey and W. Newman for sharing DKN01 validation data. We thank the St. Louis Breast Tissue Registry at Washington University School of Medicine, Department of Surgery for assistance in obtaining tissue samples and D. Veis for pathology consultation. S.L. was partially supported by the Kwanjeong Lee Chong Hwan Educational Foundation of Korea (KEF-18Am0136). J.T. was supported by F31GM146361, EE was supported by T32GM139774 and F31CA284858 and J.Y. was supported by T32CA113275 and F31CA271721-01. S.A.S. was supported by NIH grants R01 AG059244, CA217208 (S.A.S.), The U.S. Army Medical Research Acquisition Activity, 820 Chandler Street, Fort Detrick, MD 21702-5014, is the awarding and administrating acquisition office, and this was supported in part by the Office of the Assistant Secretary of Defense for Health Affairs, through the Breast Cancer Research Program, under award No. BC181712. Opinions, interpretations, conclusions, and recommendations are those of the authors and are not necessarily endorsed by the Department of Defense. This work was also supported by the Siteman Cancer Center Investment Program (NCI Cancer Center Support Grant P30CA091842), Fashion Footwear Association of New York, and the Alvin J. Siteman Cancer Center Siteman Investment Program (supported by The Foundation for Barnes-Jewish Hospital, Cancer Frontier Fund, to S.A.S.). T.A.F. was supported by P50CA171963, R01CA205239, P30CA91842, the LLS Specialized Center of Research, the Rising Tide Foundation for Cancer Research, and the Siteman Cancer Center (Team Science Award). Schema figures were created in BioRender (Faccio, R. (2025) https://BioRender.com/a54q631). Lastly, we thank Konner Wigglesworth for helping with the analyses of the human scRNAseq dataset.

## Author contributions

All authors fulfill all authorship criteria. S.L. and R.F. designed the research; S.L., B.R., J.T., E.E., Q.R., D.C., and M.S.H. performed the experiments; S.L., B.R., J.T., P.W., S.A.S., T.A.F., C.X.M., and R.F. analyzed and discussed the data; D.C., J.T., C.X.M., and T.A.F. provided human samples and S.L., D.C., and R.F. analyzed the patient data; J.Y. and J.W. analyzed scRNAseq data and S.L. and R.F. wrote the paper.

## Competing interests

The DKK1-neutralizing antibody mDKN01 and relevant IgG control antibody were provided at no cost for this study by Leap Therapeutics. No financial support from Leap Therapeutics was provided to perform this study. M.H. is employed by Leap Therapeutics and owns company stocks. None of the other authors have competing interests in this work.
