## [Transparent Peer Review file · Nature Communications]

Stroma-derived Dickkopf-1 contributes to the suppression of NK cell cytotoxicity in breast cancer

Corresponding Author: Professor Roberta Faccio

Version 0:

Reviewer comments:

Reviewer #1

(Remarks to the Author)

Lee et al. conducted a systemic or bone-specific DKK1 targeting to show reduced primary tumor growth in various breast cancer models. Authors further claim that specific deletion of CAF-derived DKK1 also limits breast cancer progression, regardless of its elevated levels in circulation and in the bone. They show that DKK1 does not support tumor proliferation directly but rather suppresses the activation and tumoricidal activity of NK cells.

Major comments:

1. Figure 2 clearly shows that DKK1 expression is cytoplasmic/nuclear in both stromal and cancer cells. This is rather surprising for an extracellular protein. While the IF needs improvement to show always the same amount of cancer vs. stromal cells, there are some critical points that need to be clarified at this step. Namely, is DKK1 truly accessible to the mDKN01 mAb? A quantitative biodistribution study should be shown in the tumor models used. An IF should be performed to show the intra-tissue localisation of the mDKN01 mAb following systemic injection. Equally important would be to examine the binding of mDKN01 mAb on fresh frozen human normal and cancerous breast (ex vivo). Co-localisation with epithelial and different stromal markers should be performed.
2. Further deriving from Figure 2, is the question of the appropriateness to use aSMA-Dkk1cKO for functional studies pertaining to CAF. One can easily see that not all PDGFRA signal coincides with aSMA signal, which is by the way normal, as aSMA is not universal to CAF. This is especially the case in breast (cancer) where myoepithelial cells in normal breast are also positive while in cancer, epithelial cells can also be positive (in addition to the ones that undergo EMT). A relevant model would have been PDGFRA driven, for example.
3. At no point do the authors show that blocking DKK1 would have signalling consequences in target cells. This needs to be examined using appropriate downstream signalling analysis, notably examining the activation of canonical and non-canonical pathways. Showing the phosphorylation of given targets would be mandatory.
4. Single cell analysis presented in the Figure 2 is a bit misleading. The data should be presented together (cancer and stromal cells), using established diagrams (such as tSNE plots). A more careful analysis of different CAF populations should be performed, notably referring to the state of the art classification that has been extensively published in the breast. Spatial transcriptomics data (that are now broadly available and accessible) should be used to corroborate the DKK1 expression profile in different cells and tumor histologies. Alternatively, a detailed RNAscope analysis should be performed.

Reviewer #2

(Remarks to the Author)

The manuscript by Lee et al. mainly focusses on the role of CAF-derived DKK-1 in immune evasion mechanisms and loss of NK cell activation of primary and bone-metastatic breast cancer. This topic is highly relevant, considering the emergence of immune-targeted therapies and emphasizing the importance of the tumor microenvironment in fostering tumor progression, which is still not fully understood. The paper reveals novel and important aspects of the role of DKK-1 in breast cancer, is well written and well structured. The performed experiments are comprehensive and include several controls to exclude misinterpretation of experiments and to underscore mechanistic findings. However, I ask the authors to address the following

comments:

1. Figure 1: The mDKN01 appears to be more efficient in the PyMT model compared to using 4T1 cells – Do you have an explanation for this? Along those lines, the latter mechanistic experiments were all performed using the ER+ PyMT, but not the triple-negative 4T1 cells, although your data from humans indicate a bulk expression of stromal DKK-1 expression in TNBCs. Can you explain this? Please discuss.
2. Figure 1: The 4T1 model was only used to assess effects of mDKN01 on primary tumor growth. Did you also test the cells in the intratibial and intracardiac model and, if so, what were the findings? Especially as human CAFs in TNBC express way more levels of DKK-1 than ER+ tumors, metastasis models using mouse TNBC cells would be of high interest.
3. Figure 2: Expression of DKK-1 in CAFs is immunohistochemically shown for the PyMT and the MMTV-PyMT model, but not for the 4T1 cells – did you check it for the 4T1 mouse model? Is there a difference of stromal DKK-1 abundance as demonstrated in Fig.2B?
4. Figure 3: Authors show that both osteoblast and fibroblast-derived DKK-1 contributes to primary tumor growth. In consideration of experiments in Figure 1, is the efficacy of mDKN01 against osteoblast- and fibroblast-derived DKK-1 similar? DKK-1 presents in different protein modifications (i.e. glycosylation) and I was wondering if the protein structure of osteoblast- and fibroblast-derived DKK-1 is similar or different and if this affects the contribution to tumor growth.
5. Figure 3: Did you see differences of NK cell activity in the osteoblast- and CAFs-specific DKK-1 depletion models?
6. Suppl. Fig.4: You only tested effects of rDKK1 on cell lines and did not observe effects on proliferation. Have you investigated DKK-1 overexpression in these cells?
7. Figure 6: You used MDA-MB-231 cells that have already high baseline levels of DKK1 and are triple-negative. Given the experiments using the ER+ PyMT model, it would be useful to assess NK cell activity in presence and absence of rDKK1 against human ER+ breast cancer cell lines, too. In addition, NK cell activity assays using tumor cell supernatants (with high vs. low DKK-1 levels) could be investigated.
8. Figure 7: Did you analyze the NK cell activity in patients with primary breast cancer and varying levels of DKK-1 but no bone metastases? The finding of reduced NK cell activity in patients with progressive bone metastases is very interesting; however, I think that should be analyzed in samples of primary breast cancer, too.
9. Figure 7: Along those lines, did you distinguish between the bone- and CAF-derived DKK-1 expression in stable vs. progressive bone metastases? Is DKK-1 in progressive bone metastases increased by bone cells or CAFs or both?
10. Figure 7: Did you observe differences in DKK-1 and NK cells in patients with progressive bone metastases present with visceral metastases compared to those with bone metastases only?
11. Figure 7: You analyzed DKK-1 and NK cell ligand expression in normal vs TNBC samples (7E) – Given that patients with bone metastases presented with ER+ tumors, it would be interesting to assess DKK-1 and NK cell ligand expression in samples from ER+ tumors, too.

Reviewer #3

(Remarks to the Author)

In this manuscript Lee and colleagues report that DKK1 suppresses NK cell activity in the context of breast cancer and in murine models clearly show that cancer-associated fibroblasts (CAF) are a critical source of DKK1.

The mouse data are compelling; however, it is unclear to what extent the findings reflect the clinical situation. This concern is born principally from the data in Figure 2.

The data in Figure 2A and B indicate that stromal and immune components express much less DKK1 than the tumor itself. Expression on epithelial (i.e. cancer) cells should be presented on the same graph as the stromal cells to allow assessment of comparative expression. In keep with the expression data, the histological data presented in Fig 2D-F suggest limited co-expression of DKK1 and stromal cell markers. Based on these data, it seems that the CAF contribution to DKK1 expression may be comparatively limited in human breast cancer compared with the contribution of cancer epithelial cells. Published data (e.g. Nat Genet 53, 1334–1347 (2021). <https://doi.org/10.1038/s41588-021-00911-1>), where DKK1 expression appears to be almost entirely in epithelial cells, is consistent with the findings presented here.

It is unclear how the human DKK1 expression data compares to expression across cell types in the mouse. From the histology in Figure 2G, it seems that non-epithelial cells express comparatively more DKK1 in mouse, with low or no expression by the cancer cells. Epithelial cell markers are not included in the mouse analyses, and this should be added.

If the cancer epithelium is the primary DKK1 expressing cell subtype in humans, the expression by CAFs may not be physiologically relevant in the human cancer setting. This can be tested in the mouse model. What happens in a mouse model if the cancer cells are the main population expressing DKK1? This may require access or generation of cancer cells

that overexpress DKK1 similar to humans. Are CAFs still relevant in this case? Previous studies have shown that DKK1 overexpression in breast cancer cells exacerbated bone metastasis (see *Nat Cell Biol* 19, 1274–1285 (2017). <https://doi.org/10.1038/ncb3613>) but did not examine the role of CAFs expressed DKK1.

The mouse data are solid and compelling and demonstrate a role for DKK1, including CAF-derived DKK1. Both the DKK1 inhibition studies and the targeted deletion models principally examined primary tumor growth (e.g. the DKK1 antibody was administered just after tumor injection) and it remains unclear whether inhibiting DKK1 later will also affect the growth of the tumor and/or metastatic spread. What happens if DKK1-neutralization is commenced after the tumor has established e.g. day 10 post tumor injection? This should be at least considered.

The targeted mouse deletion models and coinjection experiments provide evidence for the pro-tumorigenic effects of stromal-derived DKK1. Direct effects DKK1 on tumor cell growth, examined in cultures supplemented with rDKK1, indicate that DKK1 does not directly affect tumor growth but affect a series of anti-tumor pathways linked to immune responses. This is supported by analysis of immune infiltrates in mice treated with anti DKK1, with all major immune cell populations increased and present within the center of the tumor after treatment. A role for CAFs' DKK1 is supported by data generated in models where DKK1 is specifically absent in CAFs. Subsequent studies provide evidence that DKK1 alters tumor growth by suppressing the cytotoxic activity of NK cells. Here the investigators used NK cells derived from poly I:C treated mice and cultured tumor cells in the presence of rDKK1. The data are compelling however, a question remains about the impact of DKK1 on NK cells in the in vivo tumor models. Ideally NK cells isolated from the tumor beds should be used in the killing assays; this may require pooling from multiple mice. If this is not feasible, a high parameter analysis of NK cells isolated from tumor bearing mice will provide valuable data and should be performed. This can examine NK cells for cytolytic and cytokine pathways relevant to tumor control as well as expression of activating receptors (to complement the human data).

The authors should more clearly explain how they think DKK1 affects NK cells – does it alter NK cell recruitment (Fig 5A), target cell engagement (Fig 7D), cytolytic effector molecules expression (Fig 7H)? This important point is not clear.

To summarise, it is important to reconcile how the mouse model compares to the human situation. In addition to the questions raised above, the authors should comment on the sensitivity of human breast cancer cells to DKK1 and how this compares to the mouse.

The other key point relates to providing some clarity as to how DKK1 affects NK cells as discusses above.

Overall the work is of significance and the impact of fibroblast-derived DKK1 on NK cell function is novel and noteworthy.

Minor comments:

A brief explanation as to why the tumor cells are delivered intracardially or in the tibia should be included.

The images in Fig 2G and I are too small, and it is difficult to discern any of the relevant details. The distribution of DKK1 expression in tissues from PyMT (central) and MMTV-PyMT (peripheral) tumors is very distinct. Is this significant?

Reviewer #4

(Remarks to the Author)

In this manuscript, Lee et al report a direct effect of Dkk1 in NK-mediated anti-tumoral function. The authors address the source and role of local Dkk1 in the tumor microenvironment and demonstrate that high levels of DKK1 in breast cancer create an immune suppressive microenvironment through direct inhibition of NK cell tumoricidal function. Targeting DKK1 as a single agent showed significant anti-tumor effects. CAFs were observed to produce DKK1 at the tumor site but did not affect DKK1 levels systemically in contrast to bone-derived DKK1. Deletion of either source of DKK1 resulted in significant anti-tumor effects. While the data are interesting and potentially clinically relevant, they are not entirely novel. Inhibition of Dkk1 using a mDKN01 neutralizing antibody has been used in the context of metastatic breast cancer (4T1 mammary tumor cells) in Haas et al, *Mol Cancer Res* 2021, showing that Dkk1 inhibition decreases mammary tumor size and suppresses lung metastasis. This study also showed activation of NK cells upon Dkk1 inhibition in melanoma. In this study, further experiments are required to bolster experimental reproducibility and strengthen the conclusions.

Major points:

In the human samples it appears that expression of Dkk1 in tumor epithelial cells is much higher than in the stroma (max average expression of 4 vs 0.2) (Figure 2A). However, tumor cell lines do not express Dkk1 by qPCR (Suppl Fig 1D). Epithelial cells could upregulate Dkk1 in vivo during tumor progression. It would be useful to sort epithelial and stromal populations from tumors to check the levels of Dkk1.

The number of biological replicates and samples is missing throughout the manuscript, including for IHC (number of tumors in Fig 2D-F), immunofluorescence and bulk RNA-seq data.

Figure 3E: It is not clear whether littermate controls also received tamoxifen or their genotype. It is intriguing that Dkk1 KO only in aSMA-positive cells has an effect on tumor growth. However, if only aSMA-Dkk1-cKO mice received tamoxifen, the conclusions about Dkk1 derived from aSMA+ cells cannot be made. The authors exclude the possibility of tamoxifen affecting tumor growth based on in vitro studies but this should also be conducted in vivo under the same conditions as for experimental mice.

aSMA is also highly expressed in myoepithelial cells. aSMA expression in the epithelial vs stromal compartments of these mice should be assessed before assuming Dkk1 has been deleted in the fibroblast compartment. Again, Dkk1 levels are higher in epithelial cells than in stromal cells.

Other:

1. Supp Fig. 2B shows almost no expression of Dkk1 mRNA.

Please show Dkk1 expression on the uMAP for CAF subclustering. Authors also need to explain their reasoning for using these CAF markers. The data from Wu et al. state that MCAM+ clusters are perivascular-like cells and not CAFs. Is the expression in these subclusters significantly different?

2. Figure 2D: While Dkk1 expression by IF looks interesting, larger areas for IF should be used. The manuscript discusses Dkk1 expression in epithelial cells vs stroma but the areas shown are too small. The number of tumors analyzed is missing.

3. The mouse model and sorting strategy based on Tomato expression for CAFs used in Figure 3J requires further explanation, including how many cells were tomato-positive in the epithelial vs stromal compartments, etc.

4. The number of interactions evaluated for Figure 7D should be clarified, were conclusions based on n=11 interactions only?

5. Supp. Fig. 1D is described before Supp. Fig. 1B, C.

6. Figure 2H, 2J. To address co-localization, higher magnification is needed. The number of tumors analyzed is missing.

7. Figure 2C: it is unclear how these subsets were defined and why these markers were used to identify such subsets.

8. Although Dkk1 does not directly modulate tumor cell density and survival in vitro, this conclusion should be softened as it is based on in vitro data only.

9. For Fig 5B-D, FACS analysis should be performed for identification and quantification of specific immune cell types in addition to the IHC shown.

10. Supp Figure 4F, why are there mApple+CD45+ cells if these are epithelial cells?

11. Figure 4B, the Venn diagram is misleading. 16,229 genes are not the overlapping genes but the total number of genes sequenced. There is no overlap between upregulated and downregulated genes; please indicate with a different type of graph.

12. Are the pathways in Fig 4C upregulated or downregulated in mDKN01-treated mice?

13. Which tumor model is used in Figure 5A? Haas et al., 2021 should also be referenced for the increase in CD45+ and NK cells after mDKN01 treatment.

14. Since mDKN01 effects are completely abrogated in NSG mice, is the bone-specific function of Dkk1 also immune-system dependent? In the Prf1 KO model, are all the effects of Dkk1 mediated by NK cells? Is there any role for MDSC in tumor progression? (D'Amico et al., 2016)

Version 1:

Reviewer comments:

Reviewer #1

(Remarks to the Author)

The reviewer was surprised by the rather clear IF images in the new Figure 2 showing intracellular localisation for DKK1, which is supposed to be a secreted protein. In response to this comment, the Authors provided new data showing that DKK1 levels are lower in mice that have received the neutralising antibody (New Supp Fig. 1A). While the authors surprisingly take this as evidence of secreted nature of DKK1, the reviewer is not convinced. The reason is that this is a fully unexpected result for a secreted protein to be removed following the binding of the mAb to it. By which mechanism should this happen, especially if (see below) the mAb is Fc-gamma incompetent (unclear)? Actually, it is rather more expected that the levels of DKK1 should even increase with the blockade. In any case, the data in the Supp. Fig. 1A cannot be taken as evidence of the

secreted nature of DKK1 nor as evidence that the antibody is targeting DKK1 in a specific fashion. The newly included Figure 2 shows no evidence of the extracellular DKK1, none of the IF shows presence of DKK1 at the cell surface or in the ECM. Here, given the promiscuous nature of the protein (its expression in both epithelial and stromal compartments) we require more data here + confocal imaging to show the localisation (in support of this, also Figure S2, the PDGFRA stain is inside the cell, even nuclear?). Positive control tissue (uterus) and negative control (liver, hepatocytes) should be included.

The specificity of the mAb should be further proven using pull down experiment and MS. The pull down sample should be conditioned medium from CAF isolated from BC. What is the list of binders and what is the peptide coverage for each one of the them?

Biodistribution image, presented in the Rev. Fig. 1 should be included in the manuscript. The reviewer is very surprised with the data shown, because strong kidney positivity is in stark contrast to the single cell data, showing that kidney is by far the organ with very low DKK1 expression (<https://www.proteinatlas.org/ENSG00000107984-DKK1/single+cell>). Unclear is why the organs were harvested and why the imaging was not done in the living animal as usual?

Therefore the reviewer is still not convinced that the effects observed in the manuscript stem from targeting soluble DKK1. In fact the antibody used in the study is not sufficiently described. What is the backbone of the antibody IgG2a? Is it FC-gamma competent or not and if yes, how was this construction made?

In the scRNAseq, please include tSNE plot showing DKK1 expression in the entire population of cells presented in the panel A of Supp. Fig. 2.

Reviewer #2

(Remarks to the Author)

The authors have adequately addressed my comments.

Reviewer #3

(Remarks to the Author)

The authors thoughtfully addressed the points raised in my review, some of which overlapped with those raised by the other reviewers. Careful consideration of the comments, including the addition of new data, has considerably strengthened the manuscript.

One of my questions concerned providing further insight into how DKK1 influences NK cells, and the authors have supplied additional data to address this. Target cell contact (Fig. 6A) may be a direct effect of DKK1 on NK cells (although, as acknowledged by the authors, this is a bit speculative due to the difficulties in enumerating cell-cell contacts). However, as the authors show, DKK1 profoundly affects the immune landscape of the TME and therefore in vivo it is possible that DKK1 might further impact NK cell functions, including NK cell cytotoxicity, indirectly by modifying the local tumor microenvironment; this point is worthy of some discussion (note that the NK cells used in the cytotoxicity assays were isolated from spleen of poly I:C treated mice thus activated).

Minor comment: Heatmap of genes related to NK cell cytotoxicity in rDKK1 stimulated NK cells: is data available about the expression of transcripts for granzymes? If yes, it would be worthy of inclusion.

Reviewer #4

(Remarks to the Author)

The major points have been addressed satisfactorily.

Version 2:

Reviewer comments:

Reviewer #1

(Remarks to the Author)

Reviewer appreciated the effort the authors made to clarify the outstanding points. In reviewers opinion the outstanding experiment remains a clarification with respect to the specificity of the experimental antibody. In this respect, the reviewer thinks that an IP pulldown followed by proteomics analysis is useful. The fact that an antibody is in Phase 1 or even approved does not mean that this antibody is not cross reactive towards additional targets (<https://www.nature.com/articles/s41570-022-00438-x>). These phenomena should not be taken for granted. Polyspecificity is additional problem (<https://www.tandfonline.com/doi/full/10.1080/19420862.2021.1999195#d1e192>). Reviewer could not find convincing public data showing that DKK1 mAb used in this study has been checked for these potential of target phenomena.

The reviewer also requests that biodistribution data are included in the manuscript, at least as Supplemental Data. Same goes for DKK1 dosage in serum of the animals. Finally, the reviewer asks for more precise information to be provided in the

material and methods section with respect to the format of antibody, its Fc part in particular.

Point-by-point response to reviewer comments

Response to Reviewer #1

We thank Reviewer #1 for the very helpful suggestions. Responses to the reviewer comments are listed below.

1. Figure 2 clearly shows that DKK1 expression is cytoplasmic/nuclear in both stromal and cancer cells. This is rather surprising for an extracellular protein. IF needs improvement to show always the same amount of cancer vs. stromal cells

The reviewer is correct. DKK1 is a secreted protein and the neutralizing antibody, mDKN01 neutralizes extracellular DKK1 as shown by ELISA in mice receiving mDKN01 (**New Supp Fig. 1A**). We were also surprised to observe DKK1 nuclear localization in a large proportion of cancer cells, and a more diffused cytoplasmic localization in the stromal compartment. We further confirmed nuclear localization of DKK1 using in the human TNBC line MDA-MB-231 (**New Fig. 3L**). We agree with the reviewer that DKK1 nuclear localization is certainly unexpected for a secreted protein. Nuclear localization of DKK1 has been reported in colorectal cancer (PMID: 25788273). In this paper, the authors showed nuclear DKK1 at active transcriptional sites regulating several cancer-related genes, including those involved in detoxification of chemotherapeutic agents, and concluded that nuclear DKK1 is involved in chemotherapy resistance. Thus, it is possible that also in breast cancer cells nuclear DKK1 acts as a transcriptional regulator rather than a secreted factor suppressing anti-tumor immune populations. Future studies, outside the scope of this manuscript, will need to be performed to understand the role of nuclear DKK1 in breast cancer progression. To address the concerns of the original multiplex IHC images, we have included new images to better show DKK1 staining in the tumor cells and the stromal compartment from patient samples and mouse models (**New Fig. 2**).

Is DKK1 truly accessible to the mDKN01 mAb? A quantitative biodistribution study should be shown in the tumor models used. An IF should be performed to show the intra-tissue localisation of the mDKN01 mAb following systemic injection. Equally important would be to examine the binding of mDKN01 mAb on fresh frozen human normal and cancerous breast (ex vivo). Co-localisation with epithelial and different stromal markers should be performed.

We apologize for not sharing the biodistribution data of mDKN01 (**Rev Fig. 1**). For these experiments, the infrared fluorescent dye (IR)-conjugated mDKN01 or IR-conjugated IgG2a as control were delivered intravenously into C57BL/6 mice subcutaneously injected with B16 tumor cells. Similar to the PyMT model, *Dkk1* is not expressed by the B16 tumor cells but by CAFs in the primary tumor (**Rev Fig. 1A**). One hour after IR-mDKN01 or IR-IgG2a delivery, the mice were sacrificed and perfused. Liver, kidneys, lungs, spleen, brain and tumor mass were then imaged using the Licor Odyssey Clx (**Rev Fig. 1B**). IR-mDKN01 and IR-IgG2a signals were detected in the liver and spleen but not in lungs and brain. Consistent with DKK1 expression in renal tissues (PMID: 37108841), IR-mDKN01 but not IR-IgG2a signal was detected in the kidneys. Importantly, we also detected IR-mDKN01 signal in the tumor mass (**Rev Fig. 1C**, tumors from 2 animals). Since DKK1 is produced by CAFs at this site, these results indicate that the neutralizing antibody can bind to CAF-produced DKK1 in mouse models.

Unfortunately, we do not have biodistribution data in patients and we do not have access to the human DKN01 antibody to assess its binding in human tissues. However, an ongoing clinical trial in gastroesophageal cancer showed that patients with high *DKK1* tumoral expression, assessed by RNA-Scope, were more responsive to DKN01+Pembrolizumab treatment than patients with low *DKK1* tumoral expression (PMID: 33972574). The study, however, did not show DKK1 protein localization in the tumor tissues nor identified which cells expressed DKK1.

2. Further deriving from Figure 2, is the question of the appropriateness to use α SMA-Dkk1cKO for functional studies pertaining to CAF. One can easily see that not all PDGFRA signal coincides with α SMA signal, which is by the way normal, as α SMA is not universal to CAF. This is especially the case in breast (cancer) where myoepithelial cells in normal breast are also positive while in cancer, epithelial cells can also be positive (in addition to the ones that undergo EMT). A relevant model would have been PDGFRA driven, for example.

To complement the findings in the α SMA-*Dkk1cKO* mice, we generated FSP1Cre; *Dkk1^{fl/fl}* (FSP1-*Dkk1cKO*) mice to allow deletion of *Dkk1* in fibroblasts. These mice were orthotopically injected with PyMT cells and efficient deletion of *Dkk1* in the tumor mass was confirmed by qRT-PCR (New Supp Fig. 3E). Although we acknowledge that this line can also target macrophages, we did not observe any *Dkk1* expression in macrophages (Mac, Rev Fig. 2). Importantly, we observed a significant reduction of tumor

growth in FSP1-*Dkk1cKO* mice (New Supp Fig. 3D), similarly to the findings with the α SMA-*Dkk1cKO* line. We included these considerations in the manuscript (page 10 line 189).

3. At no point do the authors show that blocking DKK1 would have signaling consequences in target cells. This needs to be examined using appropriate downstream signaling analysis, notably examining the activation of canonical and non-canonical pathways. Showing the phosphorylation of given targets would be mandatory.

We included new experiments showing that recombinant DKK1 reduces key pathways required for NK cell cytotoxicity, namely the phosphorylation of AKT, ERK1/2 and ribosomal S6 protein (downstream of mTOR signaling pathway), but not STAT5, in NK cells (**New Fig. 7E**). This data is consistent with the bulk RNAseq GSEA analysis showing rDKK1-induced downregulation of PI3K/AKT/mTOR signaling pathways in NK cells. Since AKT, S6, and ERK1/2 are required for NK cell activation and cytotoxicity, these results are consistent with the inhibitory role of DKK1 on NK cell functionality. We have also analyzed AKT/ERK/S6/STAT5 phosphorylation in response to the canonical Wnt ligand Wnt3a and found no changes compared to untreated NK cells. However, rDKK1 was still sufficient to downregulate the phosphorylation of AKT/ERK/S6 in the presence of Wnt3a, suggesting that DKK1 effects are independent of canonical Wnt signaling (**New Supp Fig. 6D, E**)

4. Single cell analysis presented in the Figure 2 is a bit misleading. The data should be presented together (cancer and stromal cells), using established diagrams (such as tSNE plots). A more careful analysis of different CAF populations should be performed, notably referring to the state of the art classification that has been extensively published in the breast. Spatial transcriptomics data (that are now broadly available and accessible) should be used to corroborate the DKK1 expression profile in different cells and tumor histologies. Alternatively, a detailed RNAscope analysis should be performed.

We apologize if the original Figure 2 was misleading. The reviewer is correct and DKK1 expression in the cancer epithelial cells is much higher than in the CAFs, as shown in the scRNAseq dataset GSE176078. However, a direct comparison of *DKK1* mRNA levels in the tumor cells versus the stromal compartment could lead to data misinterpretation. As the reviewer noted in comment #1, IHC shows DKK1 nuclear staining in the cancer epithelial cells and a more diffused localization in the stromal compartment, which is a more typical distribution for a secreted protein. We further confirmed nuclear DKK1 staining in the human MDA-MB-231 TNBC line (**New Fig. 3L**), but we were unable to detect DKK1 in the tumor cell supernatant. When MDA-MB-231 cells were co-injected with *Dkk1* sufficient or deficient CAFs, we observed increased tumor growth in the presence of *Dkk1*WT CAFs compared to *Dkk1*cKO CAFs, suggesting that DKK1 is secreted by the CAFs to exert pro-tumorigenic effects (**New Fig. 3M**). Confirming this assumption, CAFs expressing DKK1 inhibited NK cell cytotoxicity, and DKK1 neutralization reversed these inhibitory effects (**New Fig. 6E**). These results indicate that CAFs release DKK1 to suppress NK cell functions.

For these reasons we modified Figure 2 to highlight:

1) Analysis of a new dataset (GSE3744) showing increased expression of DKK1 in TNBC and HER2⁺, but not in ER⁺ breast cancer, compared to healthy tissues (**New Fig. 2A, C, E**). We further confirmed higher *DKK1* expression in cancer epithelial cells from HER2⁺ and TNBC compared to ER⁺ breast cancer using the scRNAseq dataset GSE176078 (**New Supp Fig. 2C**).

2) Multiplex IHC staining of human TNBC, HER2⁺, ER⁺ patient tissue microarray (TMAs) showing DKK1 nuclear localization in the HER2⁺ and TNBC cancer epithelial cells, but little to no expression in ER⁺ breast cancer, and a more diffused localization in the stromal populations, which is in line with the typical distribution of a secreted protein (**New Fig. 2B, D, F**).

3) Analysis of CAF subsets from the scRNAseq dataset GSE176078 using state of the art classification for breast cancer (**New Supp Fig. 2E-G**) showing expression of *DKK1* in myCAFs. Multiplex IHC of human breast cancer TMAs and murine breast cancer samples further confirmed DKK1 staining in α SMA⁺ cells. We also included the analysis of a small patient dataset (GSE8977) showing increased *DKK1* in the stroma from invasive ductal carcinoma samples compared to no tumor controls (**New Fig. 2G**).

4) DKK1 staining in the stromal populations in a small cohort of DCIS patient samples who developed ipsilateral breast cancer and little to no staining in patients who did not recur after partial mastectomy (**New Fig. 2H**).

All together these results have important clinical implications as it indicates that expression of DKK1 in the tumor microenvironment together with its levels in circulation should be carefully evaluated in breast cancer patients as they could negatively affect therapies aimed at increasing anti-tumor immune responses.

Response to Reviewer #2

We thank this reviewer for considering the topic to be highly relevant, our findings novel, and that the experiments were comprehensive with appropriate controls to exclude misinterpretation. Answers to this reviewer comments are listed below.

1. Figure 1: The mDKN01 appears to be more efficient in the PyMT model compared to using 4T1 cells – Do you have an explanation for this? Along those lines, the latter mechanistic experiments were all performed using the ER⁺ PyMT, but not the triple-negative 4T1 cells, although your data from humans indicate a bulk expression of stromal DKK-1 expression in TNBCs. Can you explain this?

It is difficult to directly compare the anti-tumor effects of DKK1 neutralization using different tumor lines that grow in animals with a different genetic background. The 4T1 cell line represents a TNBC model which is notoriously more aggressive and less treatable than ER⁺ tumors. Furthermore, although PyMT represents a luminal B ER⁺ tumor model, this cell line is insensitive to estrogen deprivation, as shown by its normal growth in male mice and tamoxifen treated animals (**New Supp Fig. 3C**). Thus, we proceeded with subsequent analysis of PyMT tumor cells because all our conditional KO mouse models are in the C57BL/6 background, while the 4T1 tumor line grows in BALB/c mice. However, we have included additional experiments with the TNBC line 4T1 (see responses to suggested experiments below) confirming the relevance of DKK1 on TNBC progression.

2. Figure 1: The 4T1 model was only used to assess effects of mDKN01 on primary tumor growth. Did you also test the cells in the intratibial and intracardiac model and, if so, what were the

findings? Especially as human CAFs in TNBC express way more levels of DKK-1 than ER+ tumors, metastasis models using mouse TNBC cells would be of high interest.

Thank you for the suggestion. We performed intratibial injection of 4T1 tumor cells to study how DKK1 modulates tumor growth in the bone microenvironment. Consistent with our findings in the PyMT tumor model, we observed significant anti-tumor effects following mDKN01 treatment. Results are included in **New Fig. 1K, L**.

3. Figure 2: Expression of DKK-1 in CAFs is immunohistochemically shown for the PyMT and the MMTV-PyMT model, but not for the 4T1 cells – did you check it for the 4T1 mouse model?

We performed immunohistochemistry on 4T1 orthotopic tumors and confirmed DKK1 staining in the stroma (**New Fig. 2K, M**).

4. Figure 3: Authors show that both osteoblast and fibroblast-derived DKK-1 contributes to primary tumor growth. In consideration of experiments in Figure 1, is the efficacy of mDKN01 against osteoblast- and fibroblast-derived DKK-1 similar? DKK-1 presents in different protein modifications (i.e. glycosylation) and I was wondering if the protein structure of osteoblast- and fibroblast-derived DKK-1 is similar or different and if this affects the contribution to tumor growth. Did you see differences of NK cell activity in the osteoblast- and CAFs-specific DKK-1 depletion models?

To address this question, we performed *in vitro* co-culture experiments using CAFs isolated from the primary tumor or pre-osteoblasts isolated from the bone marrow of mice bearing primary tumors. The isolated CAFs and the pre-osteoblasts were cultured in the presence of NK cells and tumor cells +/- mDKN01 to neutralize secreted DKK1. NK cell-mediated killing of the tumor cells was assessed 4 hours later. We found that both DKK1 produced by CAFs and pre-osteoblasts suppress NK cell cytotoxicity and that adding mDKN01 rescues these effects (**New Fig. 6E, G**). These results indicate that both sources of DKK1 have immune suppressive effects and that they are neutralized by mDKN01.

6. Suppl. Fig.4: You only tested effects of rDKK1 on cell lines and did not observe effects on proliferation. Have you investigated DKK-1 overexpression in these cells?

Published data modulating expression of endogenous DKK1 in tumor lines have shown contrasting results with a reduction in lung metastases following either overexpression or knockdown of DKK1 in MDA-MB-231 and 4T1 breast cancer lines (PMID: 28892080, PMID: 35296660). Thus, we did not attempt to overexpress DKK1 to assess its endogenous effects on tumor cell proliferation. However, we thought that it would be important to address the effects of CAF-derived DKK1 on tumor progression using a tumor line that expressed DKK1. To this end, we used the MDA-MB-231 breast cancer cells, known to express DKK1, and co-injected them with either *Dkk1*WT CAFs or *Dkk1*ckO CAFs into Nude mice, which lack T cells but retain NK cells. Our new findings demonstrate a significant increase in tumor growth in mice driven by DKK1 expression in the CAFs (**New Fig. 3M**).

7. Figure 6: You used MDA-MB-231 cells that have already high baseline levels of DKK1 and are triple-negative. Given the experiments using the ER+ PyMT model, it would be useful to assess NK cell activity in presence and absence of rDKK1 against human ER+ breast cancer cell lines, too.

In addition, NK cell activity assays using tumor cell supernatants (with high vs. low DKK-1 levels) could be investigated.

We have now included the human ER⁺ breast cancer cell line T47D. Similar to the MDA-MB-231 cells, NK cells show reduced killing of T47D tumor cells in the presence of rDKK1 (**New Fig. 8C**). The amount of DKK1 secreted by the tumor cells was below the detection limit of the ELISA kits, thus making it impossible to use tumor cell supernatants as a source of different amount of DKK1.

8. Figure 7: Did you analyze the NK cell activity in patients with primary breast cancer and varying levels of DKK-1 but no bone metastases? The finding of reduced NK cell activity in patients with progressive bone metastases is very interesting; however, I think that should be analyzed in samples of primary breast cancer, too.

We agree that it would be important to address the correlation of DKK1 levels and NK cell activation in early-stage breast cancer patients. Unfortunately, at this time, we only had access to serum and PBMCs from patients with bone metastases. However, we were able to analyze DCIS tissue samples with clinical annotations and found DKK1 staining in stromal populations in 9 patients out of 10 who developed ipsilateral breast cancer (**New Fig 2H**).

9. Figure 7: Along those lines, did you distinguish between the bone- and CAF-derived DKK-1 expression in stable vs. progressive bone metastases? Is DKK-1 in progressive bone metastases increased by bone cells or CAFs or both?

Unfortunately, this question cannot be addressed with the patient samples available to us, as we do not have primary tumor nor bone biopsies from these patients. Considering that patients with bone metastases no longer have a primary tumor is more likely that DKK1 is expressed by bone cells, but we cannot exclude contribution of fibroblasts or potentially additional cell types located at other sites. Nevertheless, our findings indicate that increased DKK1 circulating levels are observed in patients with progressive disease along with a reduction in activated NK populations.

10. Figure 7: Did you observe differences in DKK-1 and NK cells in patients with progressive bone metastases present with visceral metastases compared to those with bone metastases only?

Unfortunately, we have limited number of patient samples to address this question. However, a previous study has shown that DKK1 levels are higher in patients with bone metastases as compared to those with visceral metastases (PMID: 17876334). We included this reference in the discussion (page 21, line 415).

11. Figure 7: You analyzed DKK-1 and NK cell ligand expression in normal vs TNBC samples (7E) – Given that patients with bone metastases presented with ER⁺ tumors, it would be interesting to assess DKK-1 and NK cell ligand expression in samples from ER⁺ tumors, too.

As suggested by the reviewer, we analyzed the GSE3744 dataset but did not find increased *DKK1* expression in ER⁺ tumors compared to the healthy breast tissues (**New Fig. 2E**), nor changes in NK cell ligand expression (**New Supp Fig. 7G**). However, this dataset is derived from primary tumors, and we cannot exclude changes in NK cell ligand expression in the metastatic setting. Considering our *ex vivo* data showing that DKK1 has direct effects on NK cells (**New Fig. 7**), and data from others showing that DKK1 reduces NK cell activating ligands on dormant tumor cells in

the lung (PMID: 27015306), it is possible that DKK1 has multiple immune suppressive mechanisms that vary depending on the breast cancer subtype and/or tumor stage.

Response to Reviewer #3

We thank reviewer 3 for considering the mouse data as both solid and compelling, the work of significance and the impact of fibroblast-derived DKK1 on NK cell function novel and noteworthy. Responses to the reviewer concerns related to what extent the findings reflect the clinical situation are included below.

The data in Figure 2A and B indicate that stromal and immune components express much less DKK1 than the tumor itself. Expression on epithelial (i.e. cancer) cells should be presented on the same graph as the stromal cells to allow assessment of comparative expression. In keep with the expression data, the histological data presented in Fig 2D-F suggest limited co-expression of DKK1 and stromal cell markers. Based on these data, it seems that the CAF contribution to DKK1 expression may be comparatively limited in human breast cancer compared with the contribution of cancer epithelial cells. Published data (e.g. Nat Genet 53, 1334–1347 (2021). <https://doi.org/10.1038/s41588-021-00911-1>), where DKK1 expression appears to be almost entirely in epithelial cells, is consistent with the findings presented here.

We appreciate the reviewer mentioning this concern as this was also raised by Reviewer #1. Please refer to Reviewer #1 Comment #4 for a detailed response. We would like to emphasize that by IHC and IF we find DKK1 to have a distinct nuclear localization in the human cancer epithelial cells and in the TNBC MDA-MB-231 tumor line (**New Fig. 2B, D, New Fig. 3L**). In contrast, CAFs show a more diffused localization, typical for a secreted protein, suggesting that CAF-derived DKK1 is released in the TME to exert immune suppressive effects. We have included new co-injection experiments using DKK1 expressing tumor cells and *Dkk1*-sufficient or *Dkk1*-deficient CAFs and found that co-injection with *Dkk1*-deficient CAFs reduces tumor growth compared to *Dkk1*-sufficient CAFs. These new experiments highlight the importance of CAF-derived DKK1 despite DKK1 expression in the tumor cells (**New Fig. 3M**). We included all these considerations in the discussion (page 20, line 392).

It is unclear how the human DKK1 expression data compares to expression across cell types in the mouse. From the histology in Figure 2G, it seems that non-epithelial cells express comparatively more DKK1 in mouse, with low or no expression by the cancer cells. Epithelial cell markers are not included in the mouse analyses, and this should be added.

Thank you for the suggestions. Unfortunately, the PyMT cell line does not express epithelial cell markers including EpCAM or E-cadherin and thus in this orthotopic model we were unable to stain the epithelial cells. However, in the MMTV-PyMT spontaneous model, we can detect EpCAM⁺ tumor cells (cyan). We confirmed that DKK1 is not expressed in the epithelial compartment while is detected in the stroma (green α SMA and white COL1a1; **New Fig. 2L**).

If the cancer epithelium is the primary DKK1 expressing cell subtype in humans, the expression by CAFs may not be physiologically relevant in the human cancer setting. This can be tested in

the mouse model. What happens in a mouse model if the cancer cells are the main population expressing DKK1? This may require access or generation of cancer cells that overexpress DKK1 similar to humans. Are CAFs still relevant in this case? Previous studies have shown that DKK1 overexpression in breast cancer cells exacerbated bone metastasis (see Nat Cell Biol 19, 1274–1285 (2017). <https://doi.org/10.1038/ncb3613>) but did not examine the role of CAFs expressed DKK1.

We agree that our findings did not address the relative contribution of tumor versus stroma DKK1 in the human settings. The experiment suggested by the reviewer is very clever, however, overexpression systems could lead to results difficult to interpret in case the levels of expression of the protein of interest are supraphysiological, which could mask the effects of endogenous proteins in other cell types. Indeed, studies have shown contradictory effects of manipulating DKK1 expression in breast cancer cells on lung metastasis. One study showed that DKK1 overexpression in MDA-MB-231 reduces lung metastases (PMID: 28892080) and another study showed that knock-down of endogenous DKK1 in MDA-MB-231 also reduces metastases (PMID: 35296660). To overcome this issue, we utilized a different approach. We injected MDA-MB-231 human breast cancer cells, which express relatively high levels of *DKK1* (PMID: 29772510), with either *Dkk1*WT CAFs (isolated from tumors in α SMA-*Dkk1*WT mice) or *Dkk1*cKO CAFs (isolated from tumors in α SMA-*Dkk1*cKO mice) into Nude mice. These mice lack T cells but have NK cells. We found a significant increase in tumor growth driven by expression of DKK1 in the CAFs (**New Fig. 3M**). These results highlight the importance of stroma-derived DKK1 in supporting tumor progression, even when the tumor cells express DKK1.

The mouse data are solid and compelling and demonstrate a role for DKK1, including CAF-derived DKK1. Both the DKK1 inhibition studies and the targeted deletion models principally examined primary tumor growth (e.g. the DKK1 antibody was administered just after tumor injection) and it remains unclear whether inhibiting DKK1 later will also affect the growth of the tumor and/or metastatic spread. What happens if DKK1-neutralization is commenced after the tumor has established e.g. day 10 post tumor injection? This should be at least considered. We thank the reviewer for this suggestion and have performed additional experiments addressing the effects of DKK1 neutralization starting treatment 7 days post tumor injection and found a significant reduction in tumor growth (**New Fig. 1H**).

The targeted mouse deletion models and coinjection experiments provide evidence for the pro-tumorigenic effects of stromal-derived DKK1. Direct effects DKK1 on tumor cell growth, examined in cultures supplemented with rDKK1, indicate that DKK1 does not directly affect tumor growth but affect a series of anti-tumor pathways linked to immune responses. This is supported by analysis of immune infiltrates in mice treated with anti DKK1, with all major immune cell populations increased and present within the center of the tumor after treatment. A role for CAFs' DKK1 is supported by data generated in models where DKK1 is specifically absent in CAFs. Subsequent studies provide evidence that DKK1 alters tumor growth by suppressing the cytotoxic activity of NK cells. Here the investigators used NK cells derived from poly I:C treated mice and cultured tumor cells in the presence of rDKK1. The data are compelling however, a question remains about the impact of DKK1 on NK cells in the in vivo tumor models. Ideally NK cells isolated from the tumor beds should be used in the killing assays; this may require pooling from multiple

mice. If this is not feasible, a high parameter analysis of NK cells isolated from tumor bearing mice will provide valuable data and should be performed. This can examine NK cells for cytolytic and cytokine pathways relevant to tumor control as well as expression of activating receptors (to complement the human data).

We agree with the reviewer about the importance of addressing the function of tumor infiltrating NK cells. Unfortunately, it would require about 60 mice to perform the proposed experiment due to the limited number of NK cells that can be isolated from each tumor. To mitigate this technical constraint, we analyzed NK cell activation *ex-vivo* following 4 hours stimulation with IL-12 + IL-15 or anti-NK1.1 antibody using NK cells isolated from orthotopic tumors or bone marrow from α SMA-DKK1WT and α SMA-DKK1cKO mice. Analysis of the NK cell activation, measured through degranulation (CD107a) and cytokine production (IFN γ), demonstrated increased IFN γ ⁺ and CD107a⁺ NK cells in the tumor mass of α SMA-Dkk1cKO mice compared to control animals (**New Fig. 7F, G**). However, no differences in NK cells activation were noted in the bone marrow of these two mouse genotypes where DKK1 levels remain the same. These results further highlight the local effects of CAF-derived DKK1 in suppressing NK cell responses in the tumor microenvironment. Results are also consistent with our bulk RNA seq data, which showed decreased gene expression of *Klrk1* (the gene encoding NK1.1), *Il12rb1* (the gene encoding the IL-12 receptor) and *Il15ra* (the gene encoding the IL-15 receptor) in rDKK1-stimulated NK cells (**New Fig. 7D**).

The authors should more clearly explain how they think DKK1 affects NK cells – does it alter NK cell recruitment (Fig 5A), target cell engagement (Fig 7D), cytolytic effector molecules expression (Fig 7H)? This important point is not clear.

We appreciate the reviewer bringing this to our attention. Our results suggest that DKK1 has broad effects on NK cells. The bulk RNAseq data demonstrates downregulation of cytokine receptors (IL-2, IL-12, IL-15, IL-18, IFN γ , IFN α), adhesion molecules, and functional enzymes. In addition, the signaling data, show reduced pAKT, pERK and pS6, which are required for NK maturation and activation, indicating that DKK1 could affect both the maturation and functionality of these cells. We have clarified the impact of DKK1 on NK cells in the discussion (page 19, line 381).

To summarize, it is important to reconcile how the mouse model compares to the human situation. In addition to the questions raised above, the authors should comment on the sensitivity of human breast cancer cells to DKK1 and how this compares to the mouse.

To address whether CAF-derived DKK1 affect human breast cancer cells, we co-injected MDA-MB-231 cancer cells with *Dkk1*-sufficient or *Dkk1*-deficient CAFs and found increased tumor growth driven by DKK1 expression in the CAFs (**New Fig. 3M**). Furthermore, new data shown in **New Fig. 2H** show expression of DKK1 in the stromal cells of 9 out of 10 patients with DCIS who developed ipsilateral breast cancer after partial mastectomy, but no DKK1 expression in the 3 samples from patients who did not progress. Although the number of samples analyzed was limited, these results support the need to further evaluate stromal expression of DKK1 in breast cancer initiation and progression.

The other key point relates to providing some clarity as to how DKK1 affects NK cells as discussed above.

We have modified the discussion to better explain the broad effects of DKK1 on NK cells (page 19, line 381).

Overall, the work is of significance and the impact of fibroblast-derived DKK1 on NK cell function is novel and noteworthy.

Minor comments:

A brief explanation as to why the tumor cells are delivered intracardially or in the tibia should be included.

We used intracardiac model to study metastatic dissemination to various organs including the bone and intratibial injection to study tumor growth in the bone, now included in the result session (page 5, line 92).

The images in Fig 2G and I are too small, and it is difficult to discern any of the relevant details. The distribution of DKK1 expression in tissues from PyMT (central) and MMTV-PyMT (peripheral) tumors is very distinct. Is this significant?

The distribution of DKK1 appears to be distinct because of the different amount of stroma in these two models with the spontaneous MMTV-PyMT tumors having more stromal component than the orthotopically injected PyMT tumors. We have updated the figures to address the concerns of the reviewer (**New Fig. 2I, J, L, New Supp Fig. 2I**).

Response to Reviewer #4

We thank the reviewer for finding the data interesting and potentially clinically relevant. We have included further experiments to bolster experimental reproducibility and strengthen the conclusions as suggested. Responses to the reviewer comments are listed below

Major points:

In the human samples it appears that expression of Dkk1 in tumor epithelial cells is much higher than in the stroma (max average expression of 4 vs 0.2) (Figure 2A). However, tumor cell lines do not express Dkk1 by qPCR (Suppl Fig 1D). Epithelial cells could upregulate Dkk1 in vivo during tumor progression. It would be useful to sort epithelial and stromal populations from tumors to check the levels of Dkk1.

We agree with the reviewer's comment that the scRNAseq data analysis shows dominant expression of *DKK1* in the cancer epithelial cells compared to the stromal compartment. As also mentioned in detail to Reviewer #1 in point #4 and to Reviewer #3, we find different localization of DKK1 in the tumor cells versus the stromal cells (**New Fig. 2**). In the human tissues, tumor cells show expression of DKK1 in the nucleus, which is not the expected localization for a secreted protein. Thus, even if *DKK1* transcripts are more highly expressed in the tumor cells, if DKK1 is

not secreted, it will not have the expected immune suppressive effects. As stated above, we also performed additional experiments using the human MDA-MB-231 tumor line which expresses DKK1 and assessed *in vivo* tumor growth following co-injection with *Dkk1* sufficient or deficient CAFs. Our new data indicate that DKK1 expression in CAFs is required to support tumor growth *in vivo*, regardless of DKK1 being expressed by the tumor cells (**New Fig. 3M**).

The number of biological replicates and samples is missing throughout the manuscript, including for IHC (number of tumors in Fig 2D-F), immunofluorescence and bulk RNA-seq data. We apologize for this omission. The number of samples are now included in the Figure legend.

Figure 3E: It is not clear whether littermate controls also received tamoxifen or their genotype. It is intriguing that *Dkk1* KO only in α SMA-positive cells has an effect on tumor growth. However, if only α SMA-*Dkk1*-cKO mice received tamoxifen, the conclusions about *Dkk1* derived from α SMA+ cells cannot be made. The authors exclude the possibility of tamoxifen affecting tumor growth based on *in vitro* studies but this should also be conducted *in vivo* under the same conditions as for experimental mice.

We apologize if experimental details were not clear. Tamoxifen was also added to control mice and this information is now included in the results (page 9, line 177) and methods (page 23, line 463).

α SMA is also highly expressed in myoepithelial cells. α SMA expression in the epithelial vs stromal compartments of these mice should be assessed before assuming *Dkk1* has been deleted in the fibroblast compartment. Again, *Dkk1* levels are higher in epithelial cells than in stromal cells.

We acknowledge that α SMA can be expressed by myoepithelial cells. To address this limitation, we generated a new mouse model by crossing Fibroblast-specific protein 1 (FSP1) Cre with *Dkk1* floxed mice (*Dkk1*^{fl/fl}), thereby generating the FSP1-*Dkk1*cKO mice to broadly delete DKK1 expression in fibroblasts. We confirmed efficient deletion of *Dkk1* in the tumor mass by qRT-PCR and importantly FSP1-*Dkk1*cKO mice show a significant reduction in tumor growth thus confirming the results with the α SMA-*Dkk1*cKO line (**New Supp Fig. 3D, E**, please also see response to Reviewer #1 Comment #2). We have now included these considerations in the manuscript (page 10 line 189).

Other:

1. Supp Fig. 2B shows almost no expression of *Dkk1* mRNA. Please show *Dkk1* expression on the uMAP for CAF subclustering. Authors also need to explain their reasoning for using these CAF markers. The data from Wu et al. state that MCAM+ clusters are perivascular-like cells and not CAFs. Is the expression in these subclusters significantly different?

We have done the subclustering based on the published paper (Ye et al. Cancer Discov. (2024) PMID: 38683161) and found *Dkk1* expression in myCAFs. According to Wu et al., *ACTA2*⁺*PDGFRB*⁺*MCAM*⁺ cells are clustered as perivascular-like cells, while in Ye et al. *ACTA2*⁺*PDGFRB*⁺*COL1a1*⁺*MCAM*⁺ cells are clustered as vascular CAF (vCAF). *DKK1* is barely expressed in these subsets (**New Supp Fig. 2G**).

2. Figure 2D: While Dkk1 expression by IF looks interesting, larger areas for IF should be used. The manuscript discusses Dkk1 expression in epithelial cells vs stroma but the areas shown are too small. The number of tumors analyzed is missing.

Thank you for the suggestion. We have now included better images for IHC, and the number of tumors analyzed in the Figure legend.

3. The mouse model and sorting strategy based on Tomato expression for CAFs used in Figure 3J requires further explanation, including how many cells were tomato-positive in the epithelial vs stromal compartments, etc.

Because CAFs represent a small percentage on the tumor mass, we followed the protocol from Akinjiyan et al. (PMID: 37996700). In short, we first isolated the CAFs by plating the single cell suspension of the tumor mass for 30 min on a tissue culture dish, followed by media replacement to remove immune and tumor cells in suspension. After 20 hours of culture, the adherent cells were carefully detached with accutase digestion, stained with anti-CD45 (immune cell marker) and FACS sorting was performed based on tdT expression and exclusion of CD45⁺ cells. This approach limits the number of contaminating CD45⁺tdT⁻ tumor cells and assure efficient sorting of the CAFs. This information is now included in the methods (page 29, line 576).

4. The number of interactions evaluated for Figure 7D should be clarified, were conclusions based on n=11 interactions only?

In order to assess NK and tumor cell interactions, we need to obtain reconstructed 100X magnification images from over 60 Z-stacks. The scanning of each NK-tumor cell pair requires over 30 minutes and can result in the quenching of the fluorescence, thus limiting the number of interactions that can be assessed. We modified the results and included these images to suggest the possible involvement of DKK1 in limiting the number of NK cells in close proximity with the tumor cells. If the reviewer feels that these results are overstated and should be removed, we will follow the reviewer suggestion. However, in support of a possible effect of DKK1 in restraining NK:tumor interactions, we now include new data showing that CAF-derived DKK1 reduces NK responses to anti-NK1.1 antibody stimulation, which is known to support NK cell engagement with the target cells and induce NK cell degranulation (**New Fig. 7F**). Please see also response to Reviewer #3 for more details about this point.

5. Supp. Fig. 1D is described before Supp. Fig. 1B, C.

Thank you for the suggestion. We reordered the figure legends accordingly.

6. Figure 2H, 2J. To address co-localization, higher magnification is needed. The number of tumors analyzed is missing.

Thank you for the suggestion. We have now included new images and number of tumors analyzed in **New Fig. 2**.

7. Figure 2C: it is unclear how these subsets were defined and why these markers were used to identify such subsets.

We have done the subclustering of CAFs based on the published paper (Ye et al. Cancer Discov. (2024) PMID: 38683161) and found myCAF_s to express *DKK1*. Based on these findings, α SMA was used for IHC staining (**New Supp Fig. 2E-G, New Fig. 2**).

8. Although *Dkk1* does not directly modulate tumor cell density and survival in vitro, this conclusion should be softened as it is based on in vitro data only.

We have modified the conclusion to address the reviewer's concern and refocused on the immunomodulatory effects of *DKK1* (page 11, line 214).

9. For Fig 5B-D, FACS analysis should be performed for identification and quantification of specific immune cell types in addition to the IHC shown.

We have now included FACS analysis of tumor infiltrating population in α SMA-*Dkk1*WT and cKO mice in **New Supp Fig. 5C**.

10. Supp Figure 4F, why are there mApple⁺CD45⁺ cells if these are epithelial cells?

Due to the larger size of tumor cells, immune cells are sometimes not efficiently separated from the singlet gating and mApple⁺CD45⁺ cells might represent immune cells that are directly interacting with tumor cells. Therefore, further CD45⁻ selection was performed to sort the tumor cells.

11. Figure 4B, the Venn diagram is misleading. 16,229 genes are not the overlapping genes but the total number of genes sequenced. There is no overlap between upregulated and downregulated genes; please indicate with a different type of graph.

We now included the total number of genes (16,363) and the number overlapping genes (16,229) between mDKN01 vs IgG treated tumors (**New Fig. 4B**).

12. Are the pathways in Fig 4C upregulated or downregulated in mDKN01-treated mice?

The KEGG pathway analysis we used does not differentiate between upregulated and downregulated genes (Partek Flow software algorithm). Instead, it accounts for the number of differentially expressed genes within each specific pathway, and scores them based on this difference. Consequently, the X-axis is plotted as an enrichment score rather than a p-value showing upregulation or downregulation.

13. Which tumor model is used in Figure 5A? Haas et al., 2021 should also be referenced for the increase in CD45⁺ and NK cells after mDKN01 treatment.

In Figure 5A, we used the PyMT breast cancer cell line. Haas et al. 2021 paper is cited in the discussion when referring to these results (page 19, line 379). Although Haas indicated that *DKK1* exerted anti-tumor effects in the B16-F0, a model that is NK cell dependent, their paper did not explore the mechanism by which *DKK1* affected NK cell responses. The novelty of our study consists in a detailed examination of the signaling pathways and unexplored direct inhibitory functional effects of *DKK1* on NK cells.

14. Since mDKN01 effects are completely abrogated in NSG mice, is the bone-specific function of

Dkk1 also immune-system dependent? In the Prf1 KO model, are all the effects of Dkk1 mediated by NK cells? Is there any role for MDSC in tumor progression? (D'Amico et al., 2016)

Since T cell depletion did not alter the anti-tumor effects of mDKN01 (**New Fig. 5F**) and considering results from *Prf1*^{-/-} mice where mDKN01 no longer exerted anti-tumor effects (**New Fig. 5H**), our result suggests that the effects of DKK1 are primarily mediated by NK cells in the breast cancer model used in this study. This comment is included on page 13, line 259.

Unlike our previous study (D'Amico 2016), which showed a decreased percentage of MDSCs following DKK1 neutralization in the B16-F10 and LLC tumor models, we did not find differences in both the percentage and the number of CD11b⁺Ly6C⁺ or CD11b⁺Ly6G⁺ cells following neutralization of DKK1 (**Rev Fig. 3**). Thus, it appears that DKK1-induced immune suppression mechanisms vary based on the type of tumor.

Reviewer #1 (Remarks to the Author):

The reviewer was surprised by the rather clear IF images in the new Figure 2 showing intracellular localisation for DKK1, which is supposed to be a secreted protein. In response to this comment, the Authors provided new data showing that DKK1 levels are lower in mice that have received the neutralising antibody (New Supp Fig. 1A). While the authors surprisingly take this as evidence of secreted nature of DKK1, the reviewer is not convinced.

The reason is that this is a fully unexpected result for a secreted protein to be removed following the binding of the mAb to it. By which mechanism should this happen, especially if (see below) the mAb is Fc-gamma incompetent (unclear)? Actually, it is rather more expected that the levels of DKK1 should even increase with the blockade. In any case, the data in the Supp. Fig. 1A cannot be take as evidence of the secreted nature of DKK1 nor as evidence that the antibody is targeting DKK1 in a specific fashion.

The reviewer is correct, and we apologize for the misinterpretation of the ELISA results which are most likely due to mDKN01 interfering with the R&D Systems Quantikine ELISA kit (MKK100) we used. Dr. Haas and colleagues had the same issue with the R&D Systems kit (personal communication) and used the Abcam Mouse DKK1 ELISA kit (ab197746) which was not interfered by mDKN01 and, as the reviewer anticipated, shows an increase in DKK1 in serum of mDKN01 treated mice (Supp Fig. 4B from PMID: 33443105). The data in Supp Fig 1A has been removed and results modified accordingly.

To best evaluate if the DKK1 neutralizing antibody is targeting DKK1 in a specific fashion, we turned to the very well-established effects of DKK1 on bone. DKK1 is known to induce bone loss, and its neutralization has been shown to restore bone mass (PMID: 25263522, PMID: 22723594, PMID: 21531794). Thus, we have analyzed the bone mass in age matched female no tumor bearing control mice and in mice with orthotopic PyMT breast tumors, treated with IgG or mDKN01 (Rev. Fig. 1). We found that bone mass is reduced during tumor progression and mDKN01 rescues tumor-induced bone loss. This result provides additional evidence that the increased DKK1 levels in mice bearing orthotopic PyMT mammary tumors are having biological effects which can be rescued by mDKN01. This result has not been included in the current manuscript because is part of another project on the skeletal effects of mDKN01.

Supp Fig. 4B from PMID: 33443105 showing DKK1 serum levels following different doses of mDKN01 vs controls

Rev. Fig. 1. Bone mass measured by microCT as bone volume over total volume (BV/TV) in no tumor vs mice bearing PyMT tumors in the mammary fat pad (MFP) and receiving IgG or DKN01. Analyses were performed 2 weeks post tumor inoculation.

The newly included Figure 2 shows no evidence of the extracellular DKK1, none of the IF shows presence of DKK1 at the cell surface or in the ECM. Here, given the promiscuous nature of the protein (its expression in both epithelial and stromal compartments) we require more data here + confocal imaging to show the localisation (in support of this, also Figure S2, the PDGFRA stain is inside the cell, even nuclear?). Positive control tissue (uterus) and negative control (liver, hepatocytes) should be included.

We used tumor tissue microarrays from pathology reports and performed automated multiplex IHC, which requires multiple rounds of washing and stripping to stain for different proteins and identify which cells express DKK1 in the TME, thus possibly affecting our ability to detect a weaker extracellular signal from the secreted DKK1. For each antibody used, images were given an arbitrary color with the deconvolution algorithm provided

by HALO software under the supervision of a trained pathologist, which allowed us to determine the populations expressing DKK1. The specificity and titration of the anti-DKK1 antibody were determined using

Rev. Fig. 2. Immunohistochemistry image of benign breast epithelium (A) and placenta (B) stained with DKK1 developed with DAB chromogen (brown) and hematoxylin (Blue). Multiplex immunohistochemistry of human placenta stained for DKK1 (red, C) and human terminal duct lobular unit in normal breast tissue stained for PDGFR α (blue, D) and hematoxylin (white, D).

placenta as a positive control while benign breast epithelium as a negative control (Rev. Fig. 2 A, B now New Supp Fig. 2H). Placenta was also included in the TMA to setup the threshold signal in HALO and eliminate any background noise (Rev. Fig. 2C, New Supp Fig. 2J). Antibodies used for detection of CAFs and tumor cells have been extensively validated by others for IHC, including our collaborators (PMID: 38683161).

We were able to observe a strong intracellular DKK1 staining in the placenta, with a pattern similar to previously reported study PMID: 23452984. A defined DKK1 intracellular staining by IHC was also reported in endometrial cancer (Fig. 2 from PMID: 31865530), gastric cancer (Fig. 3 from PMID: 29720122), colorectal cancer (Fig. 1, PMID: 25788273) and pancreatic cancer (Fig. 6, PMID: 26101916). We have modified the results to include more technical information about the staining, and the limitation of our approach in detecting secreted DKK1.

For the mouse studies, we have validated the anti-DKK1 antibody by using tumors isolated from Dkk1cKO mice and confirmed no DKK1 staining, a result also confirmed by qRT-PCR. Thus, we are confident of the specificity of the DKK1 staining, but we acknowledge the limitation of using IHC which impacts our ability to detect secreted DKK1.

Lastly, we have included hematoxylin staining to detect the nuclei in the PDGFR α stained human breast tissues (Rev. Fig. 2D). PDGFR α does not appear to have any specific nuclear localization and since this is a human breast tissue section, rather than cells plated on coverslips, it is possible that the apparent diffused staining is rather due to the thickness and/or orientation of the tissue.

The specificity of the mAb should be further proven using pull down experiment and MS. The pull down sample should be conditioned medium from CAF isolated from BC. What is the list of binders and what is the peptide coverage for each one of the them?

We apologize for not providing more information in the manuscript. The mDKN01 antibody used in our study was provided by Leap Therapeutics. mDKN01 has been used by others to neutralize DKK1 in multiple cancer models (PMID: 39352454, PMID: 35924447, PMID: 33168307, PMID: 33443105, PMID: 36206576) and the human DKN01 antibody is being tested in various clinical trials for gastric and endometrial cancers (NCT04363801, NCT05761951, NCT04681248). These references have been included in the results when describing the antibody. Leap therapeutics has done extensive analyses to prove the specificity of DKN01. Please see data from reported studies that are available online (PMID and link to original data are provided).

Kinetic Parameters for Mouse and Human DKN-01						
Species	ka	T(ka)	kd	T(kd)	KD	Chi ²
Human	1.73E+6	2.8E+2	4.86E-5	5.3E+2	2.80E-11	0.071
Mouse	6.60E+5	91	1.41E-5	2.7E+2	2.13E-11	0.501

Supp Fig.1 from PMID: 33443105 shows binding kinetic measurements of human and mouse DKN01.

Leap therapeutics has also presented a poster at the 2018 AACR meeting further showing the binding specificity of DKN01 to DKK1. Please see abstract *Cancer Res* (2018) 78 (13_Supplement): 1710; <https://doi.org/10.1158/1538-7445.AM2018-1710> and poster <https://www.leaptx.com/wp-content/uploads/2022/01/AACR-2018-Poster.pdf>. Please see below key data shown in AACR poster 1710 regarding DKN01 binding specificity (Table 1, Figure 1 and Figure 2 below).

Table 1: DKN-01 Binds Multiple Species of DKK1 with High Affinity

DKK1 Species	K_D (95% Confidence Interval of fit)
Human	3.3 (1.4-7.5) pM
Murine	7.0 (4.7-11) pM
Rat	8.4 (3.9-23) pM
Rabbit	17 (11-27) pM
Cynomolgus Monkey	14 (8.4-26) pM

The equilibrium dissociation constant (K_D) of DKN-01 was determined by a kinetic exclusion assay (KinExA).

Table 2: DKN-01 is Specific for DKK1

Family Member	K_D
DKK1	3.3 pM
DKK2	> 1 μ M
DKK3	> 1 μ M
DKK4	> 1 μ M

The equilibrium dissociation constant (K_D) of DKN-01 was determined by a kinetic exclusion assay (KinExA) for DKK1, DKK3 and DKK4. The K_D for DKK2 was measured by surface plasmon resonance (Biacore).

Biodistribution image, presented in the Rev. Fig. 1 should be included in the manuscript. The reviewer is very surprised with the data shown, because strong kidney positivity is in stark contrast to the single cell data, showing that kidney is by far the organ with very low DKK1 expression (<https://www.proteinatlas.org/ENSG00000107984-DKK1/single+cell>). Unclear is why the organs were harvested and why the imaging was not done in the living animal as usual?

We apologize for not including a better explanation of the biodistribution data. The kidney is where antibodies and immune complexes are processed and excreted. Due to the larger size of immune complexes versus IgGs, it takes longer to clear immune complexes, thus the signal in the kidney in the mDKN01 treated mice is expected. Organs were removed to better study the pharmacokinetics rather than mDKN01 therapeutic effects and/or DKK1 localization. Because this experiment was performed with the B16 tumor model, and we are not characterizing the pharmacokinetics of mDKN01, we propose to refer to the mDKN01 biodistribution data confirming its presence in the primary tumor as data not shown.

Therefore, the reviewer is still not convinced that the effects observed in the manuscript stem from targeting soluble DKK1. In fact the antibody used in the study is not sufficiently described. What is the backbone of the antibody IgG2a? Is it FC-gamma competent or not and if yes, how was this construction made?

We apologize for omitting the antibody information. mDKN01 was produced by grafting complementary determining regions of mDKN01 (IgG4/kappa), with minor modifications, onto a murine IgG2a/kappa construct with a D265A substitution to abrogate FcR interactions (PMID: 18941257). This information is now added to the material and method section.

In the scRNAseq, please include tSNE plot showing DKK1 expression in the entire population of cells presented in the panel A of Supp. Fig. 2.

DKK1 UMAP plot below has been added to the New Supp Fig. 2B.

We appreciate the considerations raised by the reviewer and we understand the importance of validating our results. We have used both pharmacological inhibition of DKK1 (mDKN01), as well as multiple genetic approaches to delete *Dkk1* in various stromal populations, as well as co-injections of WT-CAFs and *Dkk1*cKO-CAFs with tumor cells, and observed similar results, namely reduced tumor growth and increased NK cell responses. The true novelty of our paper relies mostly on the unrecognized role of DKK1 on NK tumoricidal effects, which we have shown *in vivo* and *in vitro* by performing multiple complementary assays.

If the reviewer feels that conclusions related to the source of DKK1 should be softened, we can address it. We have already added to the discussion that we cannot exclude tumor cells to contribute to DKK1 release in the TME in patients, and we propose to modify the title from “Stroma-derived Dickkopf-1 suppresses NK cell cytotoxicity in breast cancer” to “Stroma-derived Dickkopf-1 contributes to suppression of NK cell cytotoxicity in breast cancer”. All new changes added to the manuscript are highlighted in yellow.

Reviewer #2 (Remarks to the Author):

The authors have adequately addressed my comments.

We thank the reviewer's valuable insights and comments which enhanced the manuscript.

Reviewer #3 (Remarks to the Author):

The authors thoughtfully addressed the points raised in my review, some of which overlapped with those raised by the other reviewers. Careful consideration of the comments, including the addition of new data, has considerably strengthened the manuscript.

One of my questions concerned providing further insight into how DKK1 influences NK cells, and the authors have supplied additional data to address this. Target cell contact (Fig. 6A) may be a direct effect of DKK1 on NK cells (although, as acknowledged by the authors, this is a bit speculative due to the difficulties in enumerating cell-cell contacts). However, as the authors show, DKK1 profoundly affects the immune landscape of the TME and therefore *in vivo* it is possible that DKK1 might further impact NK cell functions, including NK cell cytotoxicity, indirectly by modifying the local tumor microenvironment; this point is worthy of some discussion (note that the NK cells used in the cytotoxicity assays were isolated from spleen of poly I:C treated mice thus activated).

We have modified the discussion to include the reviewer comment. Please see below: “We acknowledge that there are several potential mechanisms whereby DKK1 may indirectly impact NK cell responses. A prior study identified that DKK1 could induce down regulation of NKG2D and DNAM-1 ligands in a metastatic latency breast cancer model PMID: 27015306. In addition, there was down regulation of death receptors (FAS and TRAILR), that may protect against this mode of NK cell killing. This idea is further supported by evaluation of NKG2D-CAR T cells in gastric cancer, whereby DKK1 suppressed NKG2D ligands, reducing NKG2D-based recognition PMID: 37151176. Other studies have identified that DKK1 induced macrophages to become immunosuppressive, thereby inhibiting effector CD8⁺ T cells and NK cells in gastric cancer PMID: 36206576. In future studies these mechanisms will be investigated, as well more general impact on the tumor microenvironment.”

Minor comment: Heatmap of genes related to NK cell cytotoxicity in rDKK1 stimulated NK cells: is data available about the expression of transcripts for granzymes? If yes, it would be worthy of inclusion.

Although we observed a trend, due to variability and lack of a significant difference, we did not include this data.

Reviewer #4 (Remarks to the Author):

The major points have been addressed satisfactorily.

We sincerely thank the reviewer’s valuable insights and comments which enhanced the manuscript.

Reviewer #1 (Remarks to the Author):

Reviewer appreciated the effort the authors made to clarify the outstanding points. In reviewer's opinion the outstanding experiment remains a clarification with respect to the specificity of the experimental antibody. In this respect, the reviewer thinks that an IP pull-down followed by proteomics analysis is useful. The fact that an antibody is in Phase 1 or even approved does not mean that this antibody is not cross-reactive towards additional targets (<https://www.nature.com/articles/s41570-022-00438-x>). These phenomena should not be taken for granted. Polyspecificity is an additional problem (<https://www.tandfonline.com/doi/full/10.1080/19420862.2021.1999195#d1e192>). Reviewer could not find convincing public data showing that DKK1 mAb used in this study has been checked for these potential of target phenomena.

Leap Therapeutics has shared data showing that DKN01 has been proven to specifically bind to DKK1 but not to DKK2, 3 and 4 (*Cancer Res* (2018) 78 (13_Supplement): 1710; <https://doi.org/10.1158/1538-7445.AM2018-1710> and poster <https://www.leaptx.com/wp-content/uploads/2022/01/AACR-2018-Poster.pdf>), but cannot exclude the possibility that DKN01 binds to unknown targets. However, the company shared unpublished data showing that the anti-tumor effects of DKN01 are lost in mice injected with the DKK1 KO A549 lung carcinoma cell line.

Tumor Volumes in A549 WT and DKK1 KO mice

- BALB/c nude mice were inoculated with either DKK1 WT or DKK1 KO A549 cells (5M/mouse, SC).
- When tumors reached 60mm³, animals were randomized into treatment groups and dosing initiated. Animals were treated with Isotype control (IgG2a) or mDKN-01 (10mg/kg, IP, Bi-weekly).

The reviewer also requests that biodistribution data are included in the manuscript, at least as Supplemental Data.

Data added to Supplemental Figure 9.

Same goes for DKK1 dosage in serum of the animals.

We have measured DKK1 levels in tumor-bearing mice to demonstrate changes over tumor progression and in WT and cKO mouse models to demonstrate efficiency of deletion.

Finally, the reviewer asks for more precise information to be provided in the material and methods section with respect to the format of antibody, its Fc part in particular.

All information related to the mDKN01 antibody available to us is included in the manuscript.